# Allosteric nanobodies uncover a role of hippocampal mGlu2 receptor homodimers in contextual fear consolidation

Pauline Scholler[1,2], Damien Nevoltris[2,3], Dimitri de Bundel[1], Simon Bossi[4], David Moreno-Delgado[1], Xavier Rovira [1], Thor C. Møller [1], Driss El Moustaine[1], Michaël Mathieu[1], Emilie Blanc[1], Heather McLean[4], Elodie Dupuis[2], Gérard Mathis[2], Eric Trinquet[2], Hervé Daniel[4], Emmanuel Valjent[1], Daniel Baty[3], Patrick Chames [3], Philippe Rondard [1] & Jean-Philippe Pin [1]

Antibodies have enormous therapeutic and biotechnology potential. G protein-coupled receptors (GPCRs), the main targets in drug development, are of major interest in antibody development programs. Metabotropic glutamate receptors are dimeric GPCRs that can control synaptic activity in a multitude of ways. Here we identify llama nanobodies that specifically recognize mGlu2 receptors, among the eight subtypes of mGluR subunits. Among these nanobodies, DN10 and 13 are positive allosteric modulators (PAM) on homodimeric mGlu2, while DN10 displays also a significant partial agonist activity. DN10 and DN13 have no effect on mGlu2-3 and mGlu2-4 heterodimers. These PAMs enhance the inhibitory action of the orthosteric mGlu2/mGlu3 agonist, DCG-IV, at mossy fiber terminals in the CA3 region of hippocampal slices. DN13 also impairs contextual fear memory when injected in the CA3 region of hippocampal region. These data highlight the potential of developing antibodies with allosteric actions on GPCRs to better define their roles in vivo.

[1] Institut de Génomique Fonctionnelle, CNRS, INSERM, Univ. Montpellier, F-34094 Montpellier, France. [2] Cisbio Bioassays, F-30200 Codolet, France. [3] Aix Marseille Univ, CNRS, INSERM, Institut Paoli-Calmettes, CRCM, F-13009 Marseille, France. [4] CNRS UMR9197, Université Paris-Sud, Institut des Neurosciences Paris-Saclay, F-91405 Orsay, France. Pauline Scholler and Damien Nevoltris contributed equally to this work. Correspondence and requests for materials should be addressed to P.C. (email: patrick.chames@inserm.fr) or to P.R. (email: philippe.rondard@igf.cnrs.fr) or to J.-P.P. (email: jean-philippe.pin@igf.cnrs.fr)

There is growing interest in developing either activating or inactivating antibodies with therapeutic potential[1,2], but also as innovative tools to decipher the functional roles of cell surface proteins[3,4]. G protein-coupled receptors (GPCRs), that are the main targets for small therapeutic molecules, are now considered as promising targets for therapeutic antibodies[4–8]. Single domain antibodies from camelids such as llamas (nanobodies), are particularly well suited for such purposes, being more prone to recognize specific conformations of their targets[7,9,10]. Such tools have already proven their potential for pharmacological actions[7,11], structural studies[9,12], and use as biosensors[3].

In the central nervous system (CNS), glutamate, the main excitatory neurotransmitter, exerts its fast actions via ionotropic receptors, but also modulates synaptic activity via GPCRs, so called metabotropic glutamate receptors (mGluRs)[13–15]. Eight genes encoding mGluRs are found in mammalian genomes, and are classified into three groups. While group-I receptors (mGlu1 and mGlu5) are mainly post-synaptic receptors that contribute to glutamatergic synaptic responses, group-II (mGlu2 and 3) and -III (mGlu4, 6, 7, and 8) are mainly pre-synaptically located, and inhibit transmitter release at various types of synapses[13]. As such, mGluRs are considered to be interesting targets for the treatment of various brain diseases including psychiatric or neurodegenerative diseases[13,14].

Among the various mGluR subtypes, mGlu2, but also mGlu3 and 5, open new possibilities for novel antipsychotic drugs[14,16]. However studies on the roles of mGlu2 are made difficult by the limited number of specific tools. Indeed, there are no specific mGlu2 antibodies to determine their precise localization in the brain[17]. Moreover, because of the high conservation of the orthosteric glutamate binding site located in the Venus flytrap extracellular domain (VFT) of these receptors[18], only very few selective agonists have been reported[19,20]. Efforts were concentrated on the development of positive allosteric modulators (PAMs) interacting with the less conserved 7 transmembrane domains (7TM)[18]. Albeit subtype selective PAMs have been identified, a number of limitations for their development have been observed[21,22]. Although knock out lines are available[13,14], one cannot exclude compensation during development. Eventually, mGluRs, and especially mGlu2 have been reported to associate with other mGlu subunits to form heterodimers[23–25], and evidence for mGlu2-4 heterodimers in cortico-striatal and lateral perforant path terminals has recently been provided[24,26]. These observations strengthen the need for more specific tools to better characterize the functional roles of homo or heterodimeric mGluRs containing the mGlu2 subunit.

In the present study, we aimed at identifying nanobodies[27,28] that recognize specific conformations of the mGlu2 receptor. This led us to identify two nanobodies that specifically bind to the active form of the mGlu2. Accordingly, these nanobodies act as PAMs, enhancing the agonist action at mGlu2 receptors in transfected cells and in brain slices. When injected in the hippocampus, one of these nanobodies also enhances the effect of a group-II mGluR agonist in the fear-conditioning test, demonstrating their possible use to decipher the physiological role of mGlu2 receptors in the brain. These data nicely illustrate novel possibilities to develop mGlu allosteric modulators for numerous therapeutic actions, and exemplify the use of nanobodies to allosterically modulate GPCRs.

## Results

### Identification of mGlu2 selective nanobodies.
To identify nanobodies recognizing mGlu2 receptors, HEK-293 cells transiently expressing both rat and human mGlu2 were injected in llamas, and VHH (variable domain of the heavy chain of the camelid heavy-chain antibody) encoding sequences were amplified to generate a phage display library[29]. By screening the latter using a purified rat mGlu2 receptor reconstituted into nanodiscs[30], several positive clones were isolated and three of them, DN1, DN10, and DN13 were retained for analysis. FRET based binding data (Fig. 1a) revealed that all three nanobodies bind to rat mGlu2 in the presence of ambient glutamate produced by the cells, and not to any other mGluR (Fig. 1b).

Because glutamate concentration in the blood is sufficient to fully activate mGlu2 receptors, we expected that some of our identified nanobodies could preferentially bind to the active form of the receptor (Fig. 1c). Indeed, whereas DN1 displays the same affinity for the active and inactive forms of rat mGlu2 (Fig. 1d, Supplementary Table 1), DN10 and DN13 specifically bind to the active form stabilized by the orthosteric agonist, LY379268 (Fig. 1e, f, Supplementary Table 1). No binding was detected on the inactive form of the receptor in the presence of the antagonist LY341495 (Fig. 1e, f). Note that in the absence of any added ligand, and under conditions leading to very low extracellular glutamate concentrations in the assay medium through the co-transfection of the receptor with the high affinity glutamate transporter EAAC1, no binding of DN10 and DN13 to mGlu2 could be detected (Fig. 1e, f, Supplementary Fig. 1a). This is in contrast to the conditions used for the screening and first characterization of the nanobodies where binding could be detected under basal conditions, likely due to the presence of enough glutamate produced by the cells in the assay (Fig. 1b). According to these data, the specificity of DN1, DN10, and DN13 was further examined on the eight mGlu subtypes in the presence of a saturating concentration of either agonists or antagonists in cells expressing EAAC1 (Supplementary Fig. 1b, c, respectively) in transfected cells co-expressing EAAC1 and mGlu receptors (Supplementary Fig. 1d). This transporter was added to all further experiments with transfected HEK-293 cells. The specificity of DN1 and DN13 was further confirmed using fluorescent nanobodies and cell labeling. Fluorescent DN1 could label HEK-293 cells or hippocampal neurons transfected with SNAP-mGlu2, but not those expressing mGlu3 or mGlu4, respectively (Supplementary Fig. 2a, b). Moreover, labeling of SNAP-mGlu2 expressing cultured hippocampal neurons (Fig. 1g) or HEK-293 cells (Supplementary Fig. 2c) with fluorescently labeled DN13 could only be detected in the presence of the agonist, but not under basal conditions or in the presence of the antagonist. Note that in both cases, inactive SNAP-mGlu2 receptors could be labeled at the cell surface with cell impermeant fluorescent SNAP substrates[31]. Taken together, these data demonstrate that DN10 and DN13 specifically bind to the active form of mGlu2 receptor.

### DN10 and DN13 are positive allosteric modulators of mGlu2.
A possible effect of both DN10 and DN13 was first examined using a mGlu2 biosensor[32]. This sensor makes use of the large movement between the VFTs that occurs upon receptor activation, leading to an increase in distance between the N-terminal SNAP tags carried by each subunit[32]. Such movement lead to a large decrease in lanthanide-based resonance energy transfer measured in a time resolved manner (TR-FRET) (Fig. 2a). In the presence of an $EC_{20}$ concentration of LY379268, both DN10 and DN13, were found to activate the biosensor in the high nanomolar range (Fig. 2b). A similar agonist effect of both DN10 and DN13 was observed using a functional assay based on the activation of phospholipase C by this Gi/o-coupled receptor using a chimeric Gqi protein, a robust assay for Gi coupled receptors[33] (Supplementary Fig. 3a). DN13 was further demonstrated to potentiate the natural coupling of mGlu2 to the inhibition of

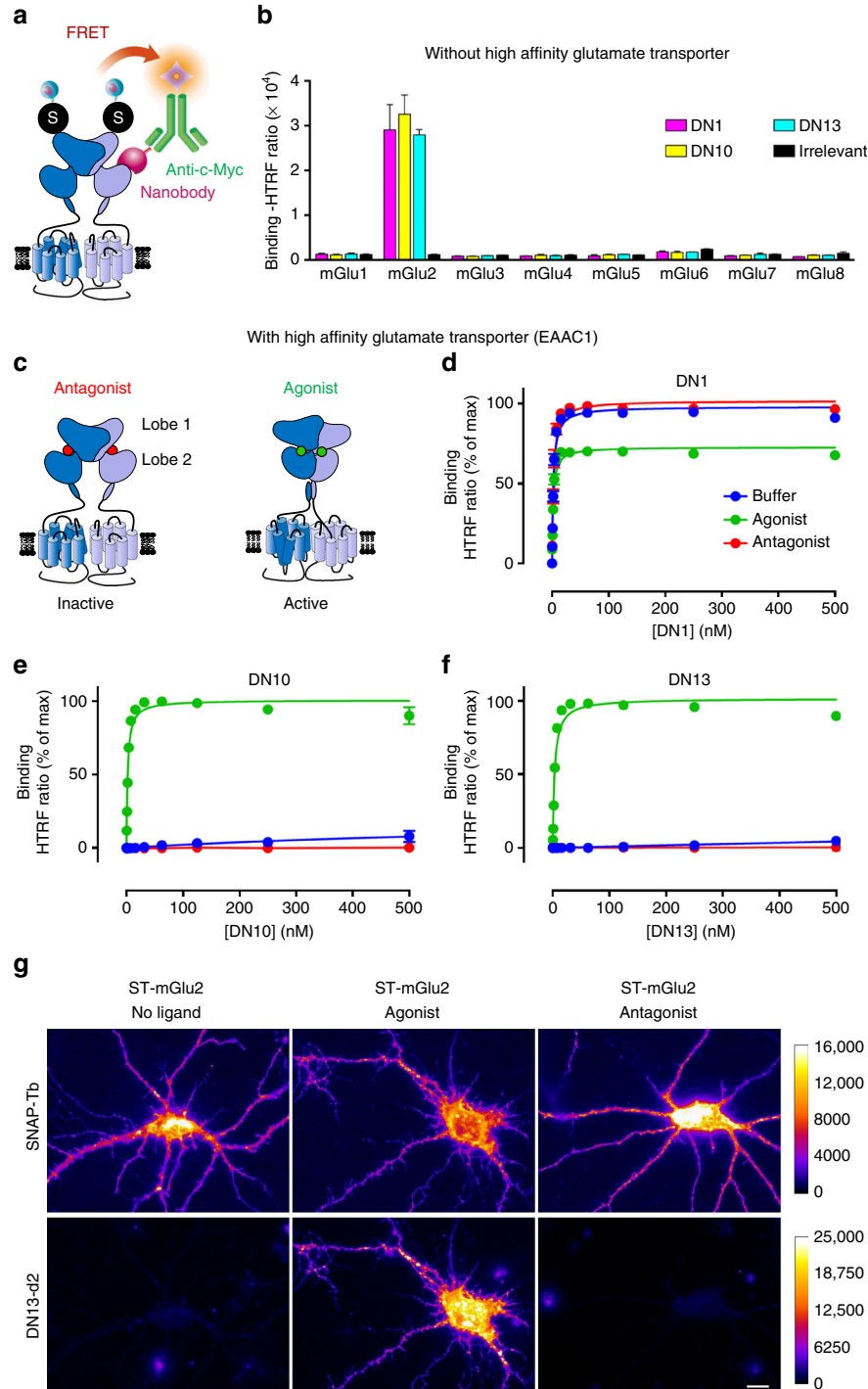

**Fig. 1** Nanobodies DN1, DN10, and DN13 specifically interact with mGlu2 receptors. **a** Cartoon illustrating the principle of the TR-FRET binding assay. The receptor fused to a SNAP-tag (black circled labeled "S") is labeled with Lumi4-Tb (light blue ball) while the nanobody (purple) bearing a c-Myc epitope at its C-terminus is labeled with 200 nM of anti-c-Myc antibody (green) coupled to d2 fluorophores (orange). Binding of the nanobody to the receptor is then measured by a TR-FRET signal. **b** Specific TR-FRET binding data obtained with the indicated mGlu receptor and either DN1, DN10, DN13, or a control irrelevant nanobody in cells under basal conditions with ambient glutamate (no high affinity glutamate transporter being co-transfected). Data are mean ± SD of triplicates from a typical experiment representative of three experiments. **c** Cartoon illustrating the main active (right), and inactive (left) conformation of an mGlu2 dimer, stabilized by an agonist (glutamate or LY379268) or the antagonist LY341495, respectively. **d–f** Saturation binding curves obtained with DN1 **d**, DN10 **e**, and DN13 **f** on mGlu2 receptors under control conditions in cells co-expressing SNAP-mGlu2 and the high affinity glutamate transporter EAAC1 (buffer), in the presence of the agonist LY379268 (1 μM), or the antagonist LY341495 (10 μM). Data are mean ± SEM of three individual experiments each performed in triplicates. **g** Fluorescence signals on cultured hippocampal neurons transfected with SNAP-mGlu2 and labeled either with SNAP-Lumi4-Tb (top) or DN13-d2 (100 nM) (bottom), under basal, agonist (LY379268, 1 μM) or antagonist (LY341495, 10 μM) conditions. Scale bar: 20 μm

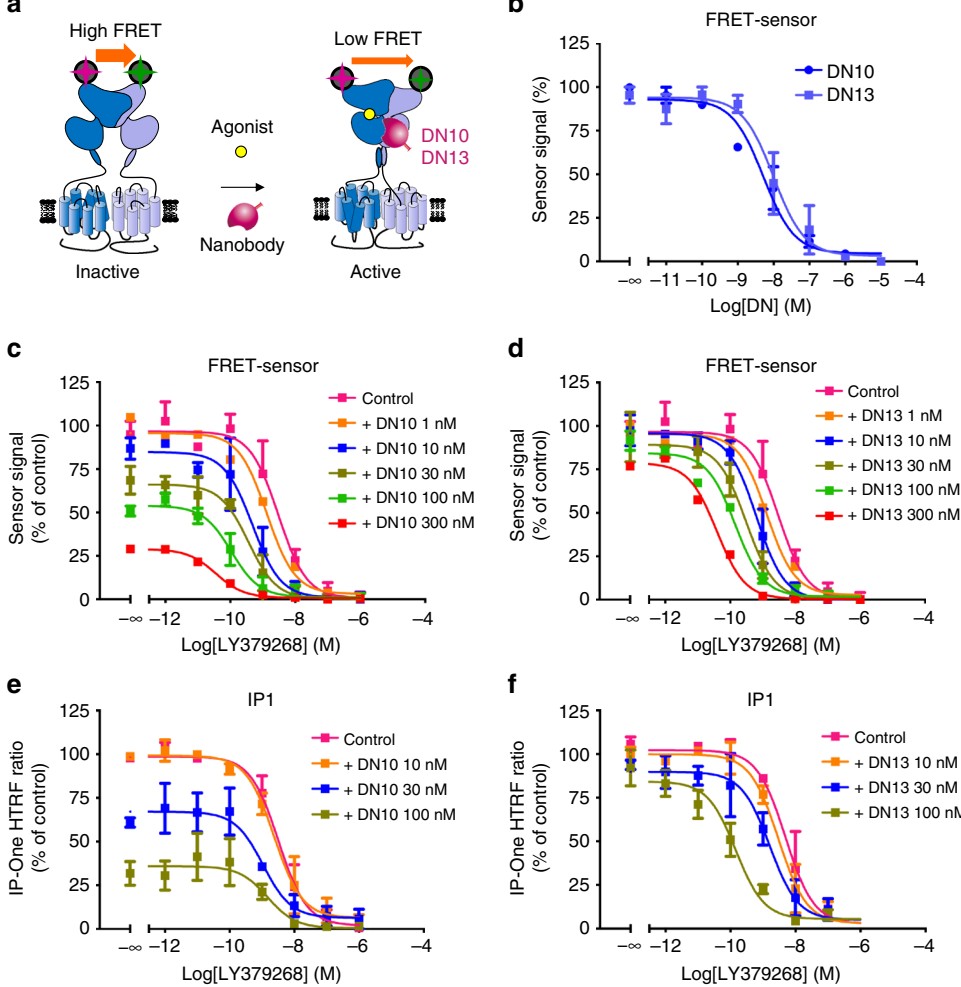

**Fig. 2** DN10 and DN13 are positive allosteric modulators of mGlu2. **a** Cartoon illustrating the principle of the mGlu2 sensor assay, where the VFT movement associated with receptor activation results in a decrease in TR-FRET signal measured between Lumi4-Tb and SNAP-Green labeled SNAP tags. **b** DN10 and DN13 dose-dependently decrease the sensor signal in the presence of $EC_{20}$ concentration of agonist LY379268 (0.1 nM), indicative of receptor activation. DN10 **c** and DN13 **d** dose-dependently potentiate the effect of LY379268 on the mGlu2 sensor. DN10 **e** and DN13 **f** dose-dependently potentiate the effect of LY379268 on the production of inositol phosphate in mGlu2-transfected cells. EAAC1 was co-transfected with mGlu2 to decrease the ambient glutamate concentration. Note that the lower the IP-One Gq HTRF ratio, the higher the amount of inositol monophosphate produced (IP1). Data are mean ± SEM of three individual experiments each performed in triplicates

cAMP formation (Supplementary Fig. 3b). When the extracellular glutamate concentration was maintained as low as possible using EAAC1, both DN10 and DN13 clearly increased mGlu2 agonist LY379268 potency (Fig. 2c, d), revealing their positive allosteric effect. However, whereas DN13 retained minimal agonist activity under such conditions (Fig. 2d), DN10 was still able to activate mGlu2 (Fig. 2c). Such observations were confirmed using the inositol-phosphate accumulation assay (Fig. 2e, f). These data demonstrate that both DN10 and DN13 act as PAMs on mGlu2, but that DN10 has, in addition, an intrinsic agonist activity (known as ago-PAM[34]).

**DN13 binds at an epitope specific of the active conformation.** The binding ability of the nanobodies on various mGlu2 constructs lacking the 7TM domains, the cysteine-rich domain, or the entire extracellular domain, revealed that all three nanobodies bind to the VFT (Supplementary Fig. 4). Moreover, competition studies revealed that DN10 and DN13 share part of their binding epitope, while DN1 binds at a different site (Supplementary Fig. 5). To identify the epitope recognized by DN13, we made use

of the inability of this nanobody to bind to the homologous mGlu3 receptor, and their lack of affinity for the inactive form of the receptor. The major conformational change in the VFT dimer occurring upon receptor activation is the relative reorientation of the VFTs leading to a close apposition of the second lobes, which are distant in the inactive form (Fig. 1c)[32]. Accordingly, a specific crevice at the interface of the second lobes is formed in the active form of the dimer only, and residues in that area were found to be different between mGlu2 and mGlu3 receptors (Fig. 3a–c, Supplementary Table 2). These include Leu226, Arg445, and Ile450 in protomer A (Fig. 3c) (Gln, Thr, and Met in mGlu3, respectively), and Ser246, Ala248, Ala249, Glu251, and Gly252 in protomer B (Ile, Lys, Ser, Asp, and Ser in mGlu3, respectively). Docking experiments conducted with a 3D model of DN13 and a mGlu2 VFT dimer model suggest that DN13 interacts at that site (Fig. 3a–c). This is further demonstrated by our observation that DN13 does not interact with a mGlu2 mutant in which these residues of the protomer B were replaced by their mGlu3 equivalent (Fig. 3d, Supplementary Table 3) despite their normal expression and coupling properties (Supplementary Fig. 6). In contrast, DN13 binds with a nanomolar affinity on

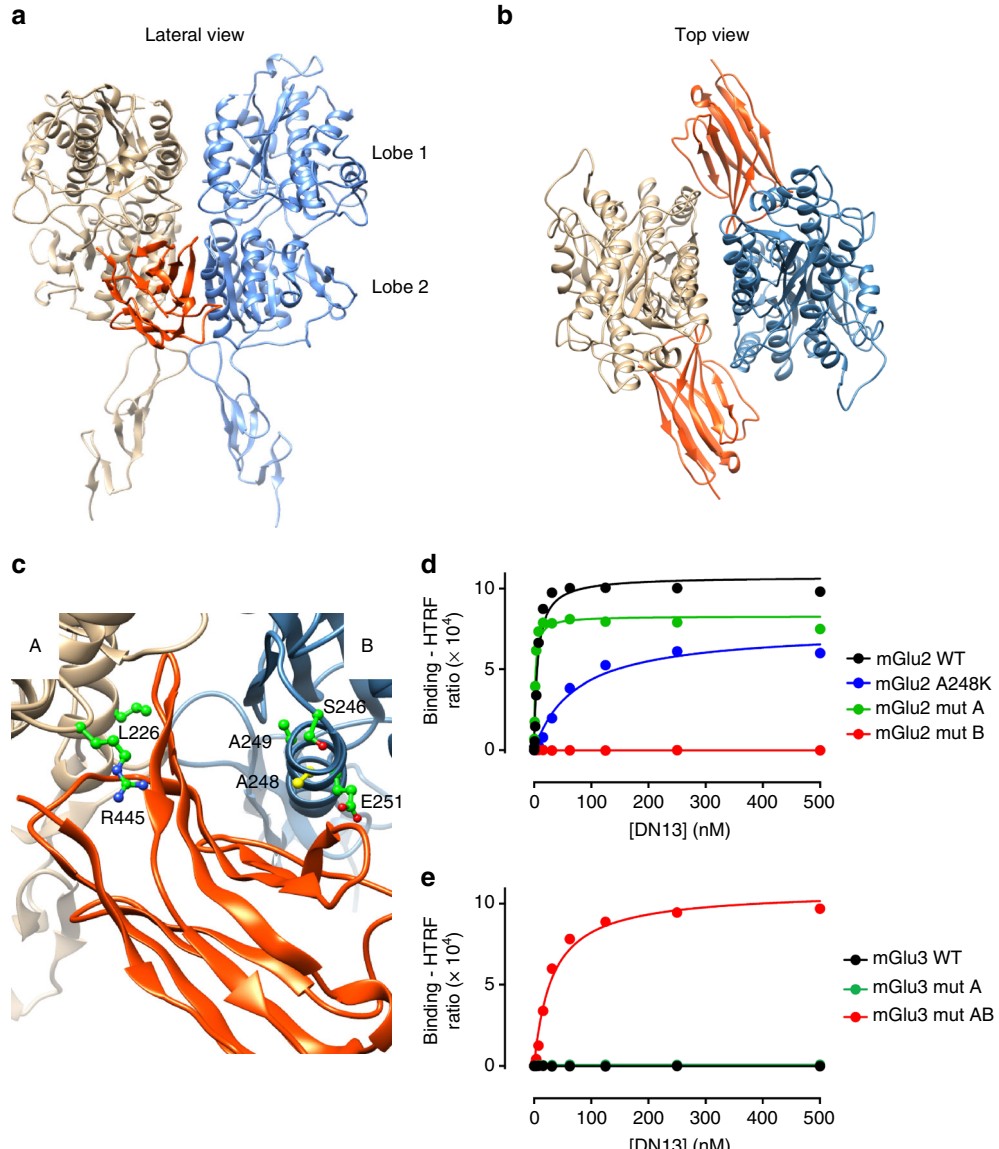

**Fig. 3** DN13 interacts at the lobe 2 crevice on the activated mGlu2 VFT dimer. **a**, **b** View of the proposed docking of DN13 (orange) on the mGlu2 extracellular domain dimer (**a**, lateral view, **b**, top view). **c** Detailed view of the proposed docking of DN13 illustrating proposed residues involved in selectivity, shown are Leu226 and Arg445 in protomer A, and Ser246, Ala248 (yellow), Ala249, and Glu251 in protomer B. **d** Saturation binding curves of DN13 on mGlu2 WT, mGlu2 bearing mGlu3 specific residues from protomer A (mut A), mGlu2 bearing mGlu3 residues from protomer B (mut B), mGlu2 A248K mutant. **e** Saturation binding curves of DN13 on mGlu3 WT, mGlu3 bearing the mGlu2 residues on protomer A (mut A), and mGlu3 bearing all identified residues of mGlu2 (mut AB) in cells co-transfected with EAAC1. Data are mean ± SD of triplicates from a typical experiment representative of three experiments

the mGlu3 mutant bearing these residues from mGlu2 (Fig. 3e, Supplementary Table 3). However, residues mutated in protomer A did not affect binding, such that more work will be necessary to identify the molecular determinants of protomer A involved in DN interaction. Our data with both mGlu2 and mGlu3 mutants show the major role played by residues in protomer B (Fig. 3d, Supplementary Table 3), and surprisingly, revealed that these residues are not critical for DN10 binding (Supplementary Table 3), suggesting that DN10 binds at a different epitope, but one that is close enough to compete with DN13 binding. As expected because of the presence of two epitopes per mGlu2 homodimer, we verified that two nanobodies can bind simultaneously to the receptor (Supplementary Fig. 7). Indeed, using d2 and Tb-labeled nanobodies, a large TR-FRET signal over background could be measured using DN1, as well as with DN10 and DN13 when the receptor is activated by an agonist.

**DN10 and DN13 do not potentiate mGlu2 heterodimers.** There is increasing evidence suggesting that the mGlu2 subunit can not only form homodimers, but can also associate with either the other group-II receptor, mGlu3, or any group-III mGluRs[23,24,26]. Because the epitope recognized by DN13 likely involves both subunits in the mGlu dimer, and DN10 binds to an overlapping area, we examined whether the nanobodies could bind to the mGlu2-3 and mGlu2-4 heterodimers by TR-FRET using d2 labeled secondary anti-c-Myc antibodies and Lumi4-Tb labeled SNAP-mGlu3 or SNAP-mGlu4, in cells co-transfected with a mGlu2 subunit (Fig. 4a). Note that the nanobodies could generate a TR-FRET signal neither on mGlu2 homodimers because of the absence of a SNAP tag on this subunit, nor on SNAP-mGlu3 or SNAP-mGlu4 homodimers because they do not bind to these receptors. Accordingly, TR-FRET signal is indicative of the binding of the nanobodies on heterodimers containing a SNAP-

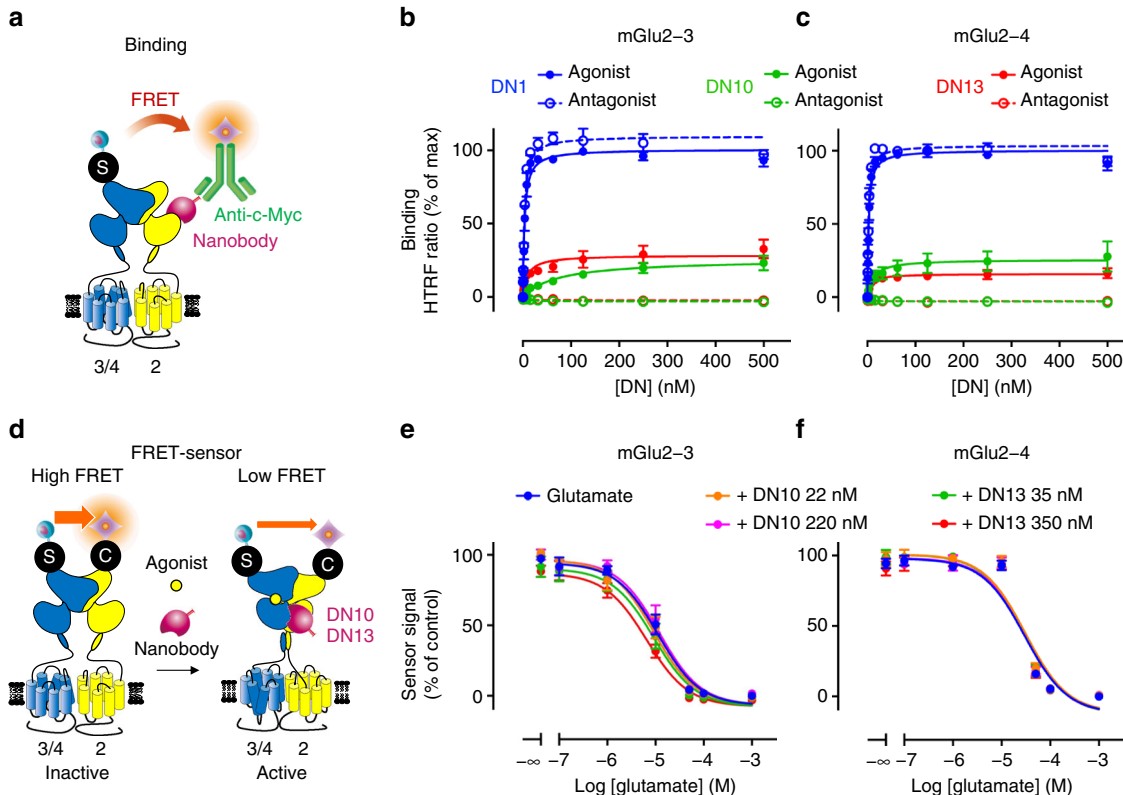

**Fig. 4** Binding and effect of DN10 and DN13 on mGlu2 containing heterodimers. **a** Scheme illustrating the approach used to detect binding of the c-Myc-tagged nanobodies on mGlu2 heterodimers containing either SNAP-mGlu3 or SNAP-mGlu4 labeled with SNAP-Lumi4-Tb. **b** and **c** TR-FRET signal measured as a function of the concentration of nanobody on cells expressing mGlu2 and SNAP-mGlu3 (**b**) or SNAP-mGlu4 (**c**). **d** Scheme illustrating the approach used to specifically detect the activation of mGlu2-3 or mGlu2-4 heterodimers, as based on the co-expression of SNAP-mGlu3 or SNAP-mGlu4 with CLIP-mGlu2, labeled respectively with SNAP-Lumi4-Tb and CLIP-Green. **e** and **f** TR-FRET signal measured as a function of the concentration of glutamate on cells expressing CLIP-mGlu2 and SNAP-mGlu3 (**e**) or SNAP-mGlu4 (**f**). Data are expressed as percent of the signal obtained with DN1 in the presence of agonist (**b**, **c**), or percent of the maximal agonist-induced change in FRET (**e**, **f**). Data are mean ± SEM of 3 independent experiments performed in triplicates

tagged subunit, either mGlu3 or mGlu4. DN1 was found to bind to both heterodimers with a similar affinity as to mGlu2 homodimers (Fig. 4b, c). In contrast, 4 times lower maximal TR-FRET signals were measured with DN10 and DN13 in the presence of saturating concentration of agonist, while no binding is detected in the presence of an antagonist (Fig. 4b, c). Of note similar TR-FRET signals are generated with these 3 nanobodies on activated mGlu2 homodimers (Fig. 1b, Supplementary Fig. 1b). Accordingly, the low saturated signals generated with DN10 and DN13 suggest that the TR-FRET originates from a small fraction of the surface receptors, and then not to the heterodimers specifically since a signal similar to that obtained with DN1 should be measured. Indeed, both mGlu2 homodimers and mGlu2 containing heterodimers are at the cell surface of the transfected cells, and receptor activation likely lead to the formation of mGlu2 containing oligomers as recently observed by our group[35]. Accordingly, it is possible that the low TR-FRET signal originates from DN interaction to mGlu2 homodimers associated with mGlu2 containing heterodimers. In agreement with this proposal, neither DN10 nor DN13 had any significant effect on the activation of both heterodimers as revealed using heterodimer specific biosensor[26] (Fig. 4d–f, Supplementary Fig. 8).

**DN10 and DN13 potentiate the pre-synaptic effect of mGlu2.** The mGlu2 gene is well expressed in hippocampal dentate gyrus granule neurons, where the receptors are targeted to mossy fiber terminals that contact pyramidal neurons in the CA3 area

(Fig. 5a). These terminals may also contain mGlu3, mGlu4, and mGlu7 that are also expressed by granule neurons[36,37]. In acute hippocampal slices, we examined the effect of nanobodies on mossy fiber terminal mGlu2 activation by quantifying presynaptic calcium transients evoked by electrical stimulation of mossy fibers, using photometric measurements of the fluorescent $Ca^{2+}$ sensitive dye, magnesium green-AM[38]. We found that saturating concentrations of DN10 (2.5 μM) progressively decrease the amplitude of $Ca^{2+}$ transients in mossy fiber terminals and slow the off rate kinetic of the inhibition produced by the group-II mGluR agonist DCG-IV applied at saturating concentration (5 μM) (Fig. 5b). This is consistent with the ago-PAM activity of this nanobody on mGlu2 homodimers. DN13 had no effect when applied alone, but also slowed down the recovery after the inhibition observed with 5 μM DCG-IV (Fig. 5c). The latter effect of DN13 involves group-II receptor activation because it can rapidly be inhibited by the group-II specific antagonist LY341495 (Supplementary Fig. 9a), and the inactive DN1 had no effect (Supplementary Fig. 9b). The PAM activity of DN13 was further confirmed through its enhancement of the inhibitory effect of low concentrations (100 nM) of DCG-IV (24 ± 1.4 and 8.3 ± 0.9% inhibition with and without DN13, respectively, $P < 0.001$, $n = 9$) (Fig. 5d). In the presence of DN13, the effect of a low concentration of DCG-IV nearly reached the maximal effect observed with saturating concentrations of the drug (30.5 ± 1.9%, $n = 6$). Taken together, these data are consistent with the DCG-IV pre-synaptic effect at the level of the mossy fiber terminals being mainly mediated by mGlu2 homodimers.

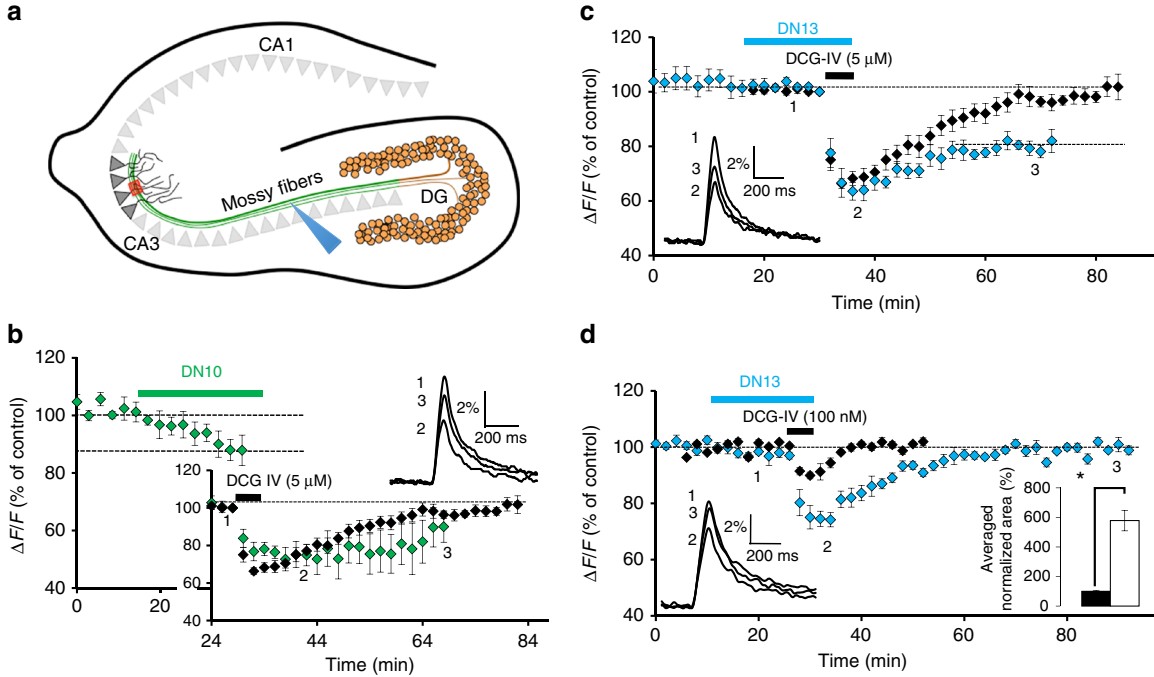

**Fig. 5** DN10 and DN13 enhance DCG-IV effect at mossy fiber terminals in the CA3 hippocampal area. **a** A schematic view of the hippocampus illustrating the granule neurons of the dentate gyrus (DG) projecting to the pyramidal neurons in the CA3 area via the mossy fibers. **b** DN10 inhibits the presynaptic calcium transients evoked by electrical stimulation of the mossy fibers (blue arrowhead in **a**) in the CA3 area (red box in **a**) and prolonged the inhibitory effect of saturating concentrations of DCG-IV (5 μM) (large inset). In this and all other panels, data are plotted as normalized amplitudes of peak fluorescence transients (ΔF/F) evoked by five stimulations of mossy fibers (delivered at 100 Hz). In the inset, data are normalized to the base line to better illustrate the difference in off rate kinetics. **c** DN13 slows the off rate kinetics of the DCG-IV (5 μM) inhibitory effect. **d** DN13 potentiates the inhibitory effect of a low concentration of DCG-IV (100 nM) on evoked presynaptic calcium transients. Right inset in **d** shows the average normalized area corresponding to the depressant effect of DCG-IV alone (black bar, $100 \pm 5.4\%$, $n = 10$) and in the presence of DN13 (white bar, $578.5 \pm 69\%$, $n = 9$), $P < 0.001$. Left inset in **c**, **d** and top inset in **b** display superimposed fluorescence changes in one of these experiments recorded at the indicated times. Each trace in **b**–**d** are an average of 10 consecutive trials. Since the variance was different between DCG-IV and DCG-IV + DN13 groups, the Welch test was applied for statistical analysis

**DN13 potentiates inhibition of contextual fear memory**. The hippocampal mossy fiber pathway projecting from the dentate gyrus to the CA3 region is critically involved in memory processing. Infusion of the group-II mGluR agonist DCG-IV into the CA3 area was previously shown to block contextual fear memory consolidation in mice[39]. Consistent with this observation we found that DCG-IV specifically disrupted contextual fear memory consolidation (Fig. 6a) when infused into the CA3 area immediately following conditioning (Fig. 6b, c and Supplementary Fig. 10) without affecting cued fear memory consolidation (Fig. 6d). Although specific for mGlu2 homodimers, we decided not to test the effect of DN10 in vivo due to its partial agonist activity, and preferred to examine only the effect of DN13. This nanobody did not affect fear memory consolidation when applied alone in the CA3 area, but potentiated the effect of low concentrations of DCG-IV on the contextual memory only, demonstrating the involvement of mGlu2 (Fig. 6c).

## Discussion

In the present study, we describe three nanobodies that specifically recognize mGlu2 in the nanomolar range, rendering them the first mGlu2 selective antibodies. Most interestingly, while one of these, DN1, does not discriminate between the different conformations of the receptor, the two others, DN10 and DN13 exclusively bind to the active form, interacting at a site only found in the active form of the dimer. Both DN13 and DN10 act as PAM, potentiating the effect of mGlu2 agonist, but DN10 also

displays a partial agonist activity. In addition we show that the use of both nanobodies can help discriminate between mGlu2 homodimers and mGlu2 containing dimers, both being inactive on both mGlu2-3 and mGlu2-4 heterodimers. DN10 and DN13 potentiated the action of mGlu2 agonists not only in heterologous expression systems, but also in brain slices and DN13 was shown to be active in vivo. These effects observed both in brain slices and in vivo at the level of the hippocampal CA3 area demonstrate the involvement of mGlu2 homodimers.

Despite the identification of both mGlu2 and mGlu3 in the early 90's, and the interest they have generated for the development of anxiolytic and antipsychotic drugs, studies of these two receptors have been hampered by the lack of specific pharmacological tools. Until now, available antibodies have not been able to discriminate between mGlu2 and mGlu3[17], and only a few selective ligands have been developed[19,20]. However, the availability of mGlu2 and mGlu3 knockout mice coupled with more sophisticated pharmacology has strengthened the interest in targeting mGlu2 specifically for antipsychotic effects[40]. Today, the most selective and promising ligands are mGlu2 PAMs that bind to a hydrophobic cavity in the 7TM[41]. Such molecules show high hydrophobicity enabling them to pass through the blood–brain barrier. However, this also limits their effective concentration in the cerebrospinal fluid, and increases the chance for off target activity[21]. Despite the therapeutic potential of PAMs, so far only orthosteric non-selective group-II mGluR agonists have reached phase 3 clinical trials for anxiety and schizophrenia[40] but have had very limited success. Our nanobodies are the first

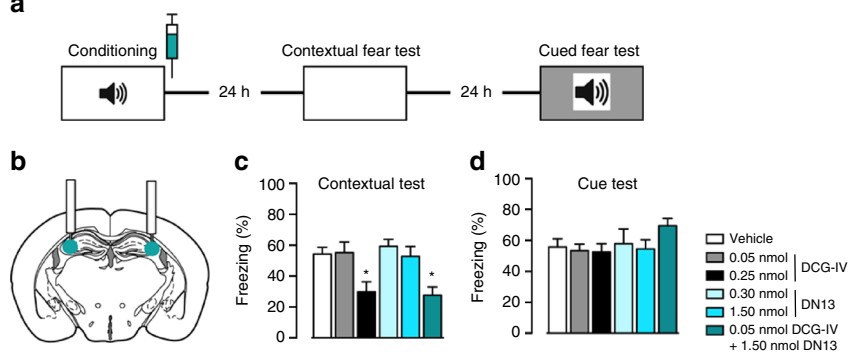

**Fig. 6** mGlu2 receptors in the CA3 area of the hippocampus control contextual fear consolidation. **a** Schematic of the experimental protocol used for the contextual fear consolidation examination in mice, and drug infusion site **b**. **c** Contextual fear memory expression. ANOVA: $F_{5,35} = 6.025$, $P = 0.0004$. **d** Cued fear memory. ANOVA: $F_{5,35} = 0.3053$, $P = 0.9066$. *$P < 0.05$ vs. PBS and **$P < 0.01$ vs. PBS

mGlu2 selective PAMs that do not target the 7TM domain, but rather the VFT domain. These molecules reveal new possibilities to develop selective mGlu2 PAMs that are designed to target this new site, without the limiting hydrophobic properties. More work will be necessary to enable them to pass the blood–brain barrier and be active after i.v. injection.

Although mGluRs were thought to exist exclusively as homo-dimers, recent data revealed that both group-I, and group-II/III mGluRs can associate to form multiple types of heterodimers[23]. Among these, the heterodimeric mGlu2-4 receptor is likely present in striato-cortical and lateral perforant path terminals, as illustrated by their specific pharmacological properties[24,26]. Much less is known about mGlu2-3 heterodimers. Our observation that DN10 and DN13 are selective for mGlu2 homodimers, over mGlu2-3 and mGlu2-4 heterodimers is in agreement with our observation that these nanobodies interact with both subunits in the mGlu2 VFT dimer. This finding will lead to multiple possibilities to develop selective nanobodies for mGlu homo and heterodimers comprised of specific mGlu subunits.

Although group-II mGluR agonists are well known for their anxiolytic and antipsychotic properties, recent data also suggest that they act to consolidate contextual fear memory[39]. This may result from a reduction in pre-synaptic glutamate release following activation of group-II mGluRs located at the mossy fiber terminals in the CA3 area of the hippocampus. These terminals originate from the granule neurons of the dentate gyrus that also express mGlu3[17] and as such, the inhibitory action of the group-II agonist DCG-IV reported at these synapses may involve either mGlu2 or mGlu3. Our data using DN13, an mGlu2 selective PAM, suggest that only mGlu2 receptors are involved in the DCG-IV effect. Eventually, the DN13-mediated potentiation of the DCG-IV effect confirmed that selective activation of mGlu2 can prevent the consolidation of contextual fear memory. Our observation that both DN10 and DN13 potentiate the action of DCG-IV at these synapses also argue against the involvement of an mGlu2 receptor heterodimer containing mGlu3 or mGlu4 subunits, and likely the other group-III mGlu7 subunit all three being expressed in hippocampal granule neurons[17,36,37]. These data argue in favor of the main involvement of mGlu2 homodimers in the DCG-IV effect at this synapse.

Taken together, our data are the first to report the development of PAM nanobodies acting at a GPCR. Antibodies show increasing potential in therapeutics, although mainly by targeting proteins other than GPCRs[1]. Since GPCRs still represent important targets for therapeutic interventions, these membrane receptors have only recently been highlighted as possible targets for antibody-based biologics[5,6]. So far, such possibilities have been validated through the identification of antibodies inhibiting

chemokine receptors[5,7]. Here we extend the use of this approach revealing the feasibility to develop nanobodies with very selective PAM activity at mGlu2 GPCRs, thus offering a way to better identify their actions in vivo, as well as localizing activated receptors within the brain. Although an access to the brain would be needed for targeting these central receptors for therapeutic intervention, conditions have been reported to facilitate brain penetration of nanobodies[42]. Moreover, mGlu receptors are not only expressed in the CNS, but also at the periphery where they have a role in the regulation of cardiovascular[43] and immune systems[44], as well as in cancer[45]. Taken together, mGluR targeting nanobodies offer interesting possibilities for therapeutic intervention.

## Methods

**Reagents, cell lines, antibodies and plasmids**. HEK293 cells (ATCC, CRL-1573, lot: 3449904) were cultivated in DMEM (Thermo Fischer Scientific, Courtaboeuf, France) complemented with 10% (v/v) fetal bovine serum. Absence of mycoplasma was routinely checked using the MycoAlert Mycoplasma detection kit (LT07-318, Lonza, Amboise, France), according to the manufacturer protocol. All drugs (DCG-IV, LY341495, LY379268, and LY487379) were from Tocris Bioscience (Bristol, UK). LSP4-2022 was provided by Dr. F. Acher (Paris, France). All HTRF reagents, labeled monoclonal antibodies anti-c-Myc-d2 (61MYCDAA, Cisbio Bioassays, Codolet, France), and anti-6His-d2 (61HISDLA, Cisbio Bioassays), labeled ligands (SNAP-Lumi4-Tb, SNAP-Red, CLIP-Red, and BG-Alexa Fluor 488) were a kind gift from Cisbio Bioassays. The pRK5 plasmids encoding wild-type rat mGluR subunits, with a HA-tag and with SNAP or CLIP inserted just after the signal peptide, were previously described[46]. pEGFP-C2 plasmid encoding EGFP was from Clontech (Mountain View, CA, USA). Point mutations were introduced in the SNAP-tag mGlu2 or mGlu3 plasmids according to the QuikChange muta-genesis protocol (Agilent Technologies, Santa Clara, CA, USA).

**Llama immunization and library construction**. Llama immunizations were executed in strict accordance with good animal practices, following the EU animal welfare legislation law and were approved by local authorities (French Ministry of Higher Education for Research and Innovation). Two llamas (Lama glama) were immunized subcutaneously four times with $5 \times 10^7$ HEK293T cells transfected with rat mGlu2 and human mGlu2 receptors. VHH library constructions were performed in E. coli TG1 strain as previously described[29,47]. Library diversities were above $10^9$ transformants.

**Selection of nanobodies by phage display**. 20 μl of the bacteria library was grown in 50 ml of 2YTAG (2YT/ampicillin 100 μg ml$^{-1}$/2% glucose) at 37 °C with shaking (250 rpm) to an $OD_{600}$ between 0.5 and 0.7. Bacteria were infected by KM13 helper phage using a multiplicity of infection of 20 during 30 min at 37 °C without shaking. The culture was centrifuged for 15 min at 3000×g, and bacterial pellet was re-suspended in 250 ml of 2YTA/kanamycin (50 μg ml$^{-1}$) for an overnight phage-nanobodies production at 30 °C with shaking. The overnight culture was split in 10 vials and centrifuged for 20 min at 3000×g. Five milliliter of 20% PEG8000, 2.5 mM NaCl were added to the supernatant in a new clean vial and incubated for 1 h on ice to induce phage particle precipitation. The solution was centrifuged for 20 min at 3000×g at 4 °C and the phage-containing pellet was re-suspended in 1 ml PBS. Another centrifugation step (2 min, 14,000×g) was performed to eliminate bacterial contaminant, and 200 μl of PEG8000 NaCl was added to supernatants in a new vial.

After 30 min on ice and a last centrifugation (5 min, 14,000×g), phage-containing pellets were re-suspended in 1 ml PBS to obtain ready to used phage-nanobodies for selections.

To obtain mGlu2 receptor specific clones, a first round of selection (S1) was performed on Maxisorp plates (Nunc, Maxisorp) coated 24 h at 4 °C with purified rat mGlu2 receptor reconstituted in nanodiscs[30]. Before selection on purified mGlu2 receptor, phage-nanobodies library was depleted by incubation with empty nanodiscs (without receptor) to eliminate anti-nanodisc antibodies and to reduce non-specific binding. Remaining phages and purified mGlu2 receptor coated wells were saturated with 2% milk/PBS during 1 h at 4 °C, and then phages and antigen were incubated together during 2 h at 4 °C for selection with shaking. Wells were then washed 10 times with PBS, and bound phages were finally eluted with 1 mg ml⁻¹ trypsine solution (Sigma-Aldrich, Saint-Quentin Fallavier, France) during 30 min at room temperature with shaking. Phages were rescued and reamplified by infection of TG1 and phage production as above, yielding S1 polyclonal phage population.

To avoid non-specific selection against proteins that composed nanodics and to select nanobodies against mGlu2 receptor in a cellular context, a second round of selection (S2) was performed on HEK293T cells transfected with rat mGlu2 receptor ($2 \times 10^7$ cells). S1 polyclonal phage population and cells were saturated in 2% milk/PBS during 1 h at 4 °C, and incubated together in the same conditions as described previously. After five PBS washes, bound phages were eluted using trypsin solution (1 mg ml⁻¹) during 30 min at room temperature. Phages were again rescued in TG1 and infected bacteria corresponding to S2 were plated. Individual TG1 colonies from S2 were picked and grown in two different 96-deep-well plates in 400 µl of 2YTAG. After overnight growth, half of the culture was frozen at −80 °C in 20% glycerol for backup, and the rest of culture was used for soluble nanobodies production induced by isopropyl-β-26-D-thiogalactopyranoside (IPTG). Nanobody concentrations in supernatant were estimated at 100–500 nM using the DoubleTag check kit (Cisbio Bioassays) according to manufacturer's recommendations.

**Production and purification of nanobodies**. For large-scale nanobody production, positive phagemids from screening step were transformed in *E. coli* BL21DE3 strain. A single colony was grown into 10 ml of LB supplemented with 100 µg ml⁻¹ ampicillin, 1% (wt/vol) glucose, and 1 mM MgCl₂ overnight at 37 °C with shaking. Then 1 l of LB supplemented with 100 µg ml⁻¹ ampicillin, 0.1% (wt/vol) glucose, and 1 mM MgCl₂ was inoculated with 10 ml of the preculture and incubated until an $OD_{600}$ of 0.7. The nanobody expression was then induced with 1 mM IPTG (final concentration) and bacteria were grown overnight at 28 °C with shaking. Bacteria were then collected by centrifugation for 10 min at 5000×g, re-suspended in 15 ml of ice-cold TES buffer (0.2 M Tris, 0.5 mM EDTA, 0.5 M sucrose, pH 8), and incubated for at least 1 h at 4 °C on a shaking platform. Twenty milliliter of TES/4 buffer (TES buffer diluted 4 times in water) were then added to the solution and further incubated for at least 45 min at 4 °C on a shaking platform. The periplasmic extract was recovered by collecting the supernatant after centrifugation of the suspension for 30 min at 10,000×g at 4 °C. The His-tagged nanobodies were then purified from the periplasmic extract by using Ni-NTA purification (Qiagen, Hilden, Germany) according to the manufacturer's instructions.

**Nanobody labeling**. Nanobodies were dialyzed overnight at 4 °C and incubated (250 µg of nanobodies at 2 mg ml⁻¹) with the fluorophore-NHS (d2-NHS (Cisbio Bioassays, Codolet, France) into carbonate buffer (15 mM Na₂CO₃, 35 mM NaHCO₃, pH 9) and Lumi4-Tb-NHS (Cisbio Bioassays, Codolet, France) in phosphate buffer 50 mM at pH 8 at a molar ratio of 6, for 45 min at room temperature. Nanobodies were then purified by gel filtration column (NAP-5) in phosphate buffer 100 mM pH 7. The final molar ratio (corresponding to the number of fluorophore per nanobodies) was calculated as the fluorophore concentration/conjugated nanobody concentration, and the conditions set up for a ratio between 2 and 3. The concentration of fluorophores in the labeled fraction was calculated as the OD/ε for each fluorophore (OD at 340 nm and $\varepsilon = 26{,}000$ M⁻¹ cm⁻¹ for Lumi4-Tb, and OD at 650 nm and $\varepsilon = 225{,}000$ M⁻¹ cm⁻¹ for d2), while that of nanobodies was determined by the $OD_{280}$. The conjugated concentration calculated as $OD_{280} - (OD_{fluo}/R_z max)/\varepsilon$ nanobody, with $R_z$ max $= OD_{fluo}/OD_{280}$. Purified labeled fractions were supplemented with 0.1% BSA and kept at −20 °C.

**Binding experiments by TR-FRET**. HEK-293 cells were cotransfected with rat SNAP-tagged mGluR and EAAC1 by electroporation (unless otherwise indicated) as previously described[23]. 24 h after transfection in a 96-well plate, cells were labeled with 300 nM SNAP-Lumi4-Tb in DMEM-GlutaMAX (Thermo Fischer Scientific) for 2 h at 37 °C, and then washed three times with Krebs buffer (10 mM Hepes pH 7.4, 146 mM NaCl, 4.2 mM KCl, 1 mM CaCl₂, 0.5 mM MgCl₂, 5.6 mM glucose, and bovine serum albumin 0.1%). Then, depending which property of the nanobodies that was investigated (affinity for the mGlu2 homo- and heterodimers, conformational selectivity, epitope mapping, and competition), different compounds (mGlu ligands, nanobodies, and anti c-Myc antibodies) were applied on labeled cells, and incubated for overnight (unless otherwise indicated) at ambient temperature.

For conformational selectivity, binding experiments were performed sequentially in basal conditions, or in the presence of antagonist or agonist in the same 96-well plate without washing step between the different conditions. First, 100 nM c-Myc-tagged nanobodies and 200 nM anti c-Myc antibodies labeled with d2 were added. Then, 2 h later and after reading of the FRET signal between Lumi4-Tb and d2 in the basal condition, the antagonist LY341495 (0.5 µM for group-I mGluRs and 10 µM for the other mGluRs) was added. Finally, after 2 h incubation and reading of this FRET signal in the antagonist condition, agonists were added (100 µM quisqualate for group-I mGluRs, 10 µM LY379268 for group-II mGluRs and 1 mM LSP4-2022 for group-III mGluR) and then FRET signal was determined. For the affinity determination, c-Myc-tagged nanobodies and 200 nM anti-c-Myc-d2 were incubated in Krebs buffer in presence of 10 µM LY341495 or 1 µM LY379268. In competition experiments, nanobodies labeled with d2 were incubated with 1 µM LY379268 and with unlabeled nanobodies. For epitope mapping, cells were transfected with Lipofectamine 2000 (Thermo Fischer Scientific) according to the manufacturer's instructions. 24 h after transfection, cells were labeled in 150 mm cell culture plate with 300 nM SNAP-Lumi4-Tb as described above, then transferred in a white 384SV-well plates (Greiner; 20,000 cells/well) using a cell dissociation buffer (Thermo Fischer Scientific). c-Myc-tagged nanobodies were incubated with 1 µM LY379268 and revealed by 200 nM of anti-c-Myc-d2. For the binding experiments on the mGlu heterodimers, cells were cotransfected with rat SNAP-tagged mGlu3 or mGlu4, HA-tagged mGlu2 and EAAC1 by electroporation and labeled with 300 nM SNAP-Lumi4-Tb as described above. c-Myc-tagged nanobodies and 200 nM anti-c-Myc-d2 were incubated with 100 µM LY341495 or 100 µM glutamate.

FRET signal was determined by measuring the sensitized d2 acceptor emission (665 nm) and Tb donor emission (620 nm) using a 50 µs delay and a 450 µs integration upon excitation at 337 nm on a PHERAstar FS (BMG LabTech, Ortenberg, Germany). TR-FRET (or HTRF) ratio (665 nm/620 nm × 10⁴, Cisbio Bioassays patent US5,527,684) was calculated for preventing interference due to medium variability and chemical compound or to normalize experiments when using cells expressing different receptors levels[48].

**Measurement of conformational change by TR-FRET**. For the mGlu2 TR-FRET biosensor, cells were transfected with rat SNAP-tagged mGlu2 and EAAC1 by electroporation and plated in a 96-well plate, as described above. 24 h after transfection, the SNAP-tagged mGlu2 homodimer was labeled with 100 nM SNAP-Lumi4-Tb and 60 nM SNAP-Green substrates for 2 h at 37 °C and then washed three times with Krebs buffer, as previously reported[32,46]. LY379268 was incubated with different concentrations of nanobodies for 1 h at ambient temperature. TR-FRET measurements were performed on a PHERAstar FS microplate reader as recently reported[26,46]. For the mGlu heterodimer TR-FRET biosensors, cells were transfected with rat SNAP-tagged mGlu3 or mGlu4, CLIP-tagged mGlu2 and EAAC1 by electroporation. 24 h after transfection, cells were labeled with 300 nM SNAP-Lumi4-Tb and 1 µM CLIP-Green in DMEM-GlutaMAX for 2 h at 37 °C, and then washed three times with Krebs buffer. Glutamate and nanobodies were incubated together. TR-FRET acceptor ratios were measured as described previously[26,46].

**Measurements of inositol phosphate and cAMP**. Phospholipase C activation was quantified by measuring the inositol monophosphate accumulation in HEK293 cells transiently expressing the mGlu receptors and a chimeric Gqi₉ protein 24 h after transfection with Lipofectamine 2000, as previously described[35]. Cells were incubated with the indicated ligands and 10 mM LiCl for 30 min, and the IP1 accumulated was quantified using the IP One HTRF assay kit from Cisbio according to the manufacturer instruction in 384-well plates[49]. The amount of cAMP was determined using the Glosensor cAMP assay (Promega Corporation, Madison, USA), as previously described[26]. HEK293 cells were co-transfected with the indicated mGluR plasmids, the pGloSensor-22F plasmid and EAAC1 encoding plasmid. The day after, cells were starved for 2 h in serum-free medium and then incubated in Krebs buffer with 450 µg ml⁻¹ luciferin (Sigma-Aldrich) for 30 min, followed by a 30 min incubation with the nanobody at the indicated concentration. The luminescence peak signal was measured on a Mithras microplate reader at 28 °C during 8 min until luminescence signal was stable. Then, forskolin (1 µM) and the indicated concentration of mGlu2 agonist LY379268 were added and luminescence was measured for 30 min.

**Cultured hippocampal neurons**. Hippocampi from Sprague-Dawley rat embryos (Janvier Labs, Saint Berthevin, France) on embryonic day 18 (E18) were dissected, dissociated by treatment with liberase TL (Roche, Basel, Switzerland) and mechanical trituration and plated on Nunc Lab-Tek II chambered cover slides (Thermo Fisher Scientific, Boston, MA, USA) coated with poly-L-ornithine and laminin at a density of ~300 neurons/mm². Neurons were cultured in Neurobasal medium (Thermo Fisher Scientific) supplemented with 2% B-27 (Thermo Fisher Scientific), 100 U ml⁻¹ Penicillin–Streptomycin (Thermo Fisher Scientific), 10 mM HEPES, and 0.5 mM GlutaMAX (Thermo Fisher Scientific). 0.5 mM L-glutamine was added when plating the cells. Half of the medium was exchanged weekly. Neurons were transfected with Lipofectamine 2000 3–6 days before imaging and imaged at 16–18 days in vitro. The medium was exchanged after 4 h of incubation

with the transfection reagent with 50% fresh medium and 50% medium conditioned by incubation with primary neurons.

**Cell imaging**. HEK293 cells were transfected with a reverse transfection protocol using Lipofectamine 2000 and plated in poly-L-ornithine coated 8-well Lab-Tek II chambered cover slides at 430 cells/mm² 24 h before imaging. Cells were co-transfected with plasmids encoding HA-ST-mGlu2 (100 ng/well) or HA-ST-mGlu3 (150 ng/well) and EAAC1 (100 ng/well). Neurons cultured in 8-well Lab-Tek II were co-transfected with plasmids encoding HA-ST-mGlu2 or HA-ST-mGlu4 (240 ng/well) and EGFP (60 ng/well, Clontech, Mountain View, CA, USA).

For DN1 experiments, HEK293 cells were labeled with 10 nM DN1-Lumi4-Tb and neurons were labeled with 100 nM DN1-d2 and 300 nM SNAP-Lumi4-Tb or 300 nM BG-Alexa Fluor 488. For DN13 experiments, cells were labeled with 100 nM DN13-d2 and 300 nM SNAP-Lumi4-Tb. DN13 labeling and imaging was done in the presence or absence of ligands: 1 µM (HEK293) or 150 nM (neurons) LY379268 or 10 µM (HEK293) or 1 µM (neurons) LY341495. Cell nuclei were stained with 2 µg ml⁻¹ Hoechst 33342 (Thermo Fisher Scientific) together with DN1/DN13 labeling. Labeling was for 1 h at 37 °C and cells were washed four times immediately before imaging. Labeling and the first wash was done in imaging buffer (127 mM NaCl, 2.8 mM KCl, 1.1 mM MgCl₂, 1.15 mM CaCl₂, 10 mM D-glucose, 10 mM HEPES pH 7.3), supplemented with 1% BSA. The remaining washes and subsequent imaging was in imaging buffer.

Images were acquired in imaging buffer with a homebuilt fluorescence microscope[50] equipped with a Fluar ×40, 1.3 NA oil immersion objective or an α Plan-Apochromat ×63, 1.46 NA oil immersion objective (Carl Zeiss Microscopy, Jena, Germany) in epifluorescence mode. Hoechst 33342, Alexa Fluor 488, and d2 were excited with a mercury lamp using the ET-DAPI, ET-GFP, and ET-Cy5 filter sets, respectively (Chroma Technology Corporation, Bellows Falls, VT, USA). Lumi4-Tb was excited with a 349 nm Nd:YLF pulsed laser at 300 Hz with ~68 µJ/pulse and the emission was collected for 3 ms after a 10 µs delay using a 550/32 bandpass filter. 4000 acquisitions were made per image. Images were shading corrected by dividing the raw image with a background image generated for each image using the 'Subtract background' function in ImageJ.

**In silico analysis of DN13 binding site**. The homology models of DN13 nanobody and the extracellular domain of mGlu₂ were generated with Modeller 9.12[51] based on the crystal structure of β₂-adrenoceptor bound nanobody (PDB 3P0G) and the mGlu₃ amino terminal domain as a template (PDB Code 2E4U)[52], respectively, using the loop optimization method. The sequences of template and modeled proteins were aligned with ClustalW2[53]. From 100 models generated, the top ten classified by DOPE score[54] were visually inspected and the best scored structure with suitable loops was chosen. The closed-closed mGlu₂ dimeric state was constructed by superimposition with the crystal structure of the active state of the extracellular domain of mGlu₁ (PDB code 1ISR)[55]. A comparison with the very recently published structure of mGlu₂ in active state (PDB code 4XAS)[56] demonstrates a close similarity with a Cα RMSD of 1.36 for the dimer and 0.86 for the monomer. The maximum structural divergence is found in the loops whereas the parts analyzed in the mutational study are very accurately located in the model.

A docking based approach was used to find the binding site of DN13 nanobody in mGlu2 according to a previously described methodology[57]. Briefly, ZDOCK 3.0 program[58] was used to perform an exhaustive rigid-body search in the six-dimensional rotational and translational space. The three rotational angles were sampled with 6° spacing, and the three translational degrees of freedom were sampled with a 1.2 Å spacing. For each set of rotational angles, only the best translationally sampled prediction was retained resulting in 54,000 predictions. The 2000 first ranked predictions were clustered with MMTSB Tool Set[59] using K-means and a radius of 2.5 Å. The 10 most populated clustered were visually inspected to avoid structural violations and symmetric results. Discovery Studio 4.0 (BIOVIA—A Dassault Systèmes brand—5005 Wateridge Vista Drive, San Diego, CA 92121 USA) was used for protein structure visualization and PDB file editing purposes. Images were generated with UCSF Chimera software[60]. The multiple sequence alignment visualization and analysis was performed with Jalview2 software[61].

**Slice preparation and calcium transient recordings**. Experiments were performed using hippocampal slices prepared from twenty six 21–25 day-old male Sprague-Dawley rats. No experiments were excluded from the analysis. In accordance with guidelines from the Centre National de la Recherche Scientifique (CNRS, France), animals were killed by decapitation after anesthesia with 2-bromo-2-chloro-1,1,1-trifluoroethan, and the brain was removed rapidly and put in an ice-cold cutting solution (75 mM sucrose, 25 mM glucose, 25 mM NaHCO₃, 2.5 mM KCl, 87 mM NaCl, 1.25 mM KH₂PO₄, 7 mM MgCl₂, and 0.5 mM CaCl₂). Parasagittal hippocampal slices, 350 µm thick, were prepared using a Vibroslicer (Motorised Advance Vibroslice MA752, Campden Instruments) according to[62]. Slices were then placed in oxygenated (saturated with 95% O₂ and 5% CO₂) artificial CSF (138.6 mM NaCl, 3 mM KCl, 1.15 mM KH₂PO₄, 1.15 mM MgSO₄, 24 mM NaHCO₃, 2 mM CaCl₂, 10 mM glucose) and left to recover at room temperature for at least 1 h. Slices were then transferred to the recording chamber,

where they were maintained at 29–30 °C and perfused with oxygenated artificial CSF as above.

Presynaptic calcium transients were recorded by photometry according to Regehr and collaborators[38,63]. A solution of 100 µM of the membrane-permeant calcium dye Magnesium Green-AM was delivered during 40 min at the level of the stratum lucidum in CA3, where mossy fibers contact proximal dendrites from pyramidal cells. After loading of the mossy fiber, slices were left for at least 30 min to allow diffusion of the fluorochrome in the fibers. A stimulation electrode filled with artificial CSF was placed between the fluorochrome loading site and the measurement window. To measure the effect of mGlu2 ligands on evoked presynaptique calcium influx, a train of five 100 Hz stimulations is delivered every 30 s to the mossy fiber to induce presynaptic calcium transients, in the presence or absence of the indicated drugs or nanobodies. During acquisition, a GABA_A receptor antagonist (bicuculline methiodide) was added to the artificial CSF to block any GABAergic component.

Measurements of intracellular calcium variations were performed on an epifluorescence microscope (Zeiss axioskop 2), with a mercury lamp (Mercury short Arc HBO, 103 W) for excitation (485 nm excitation filter), and a 530 nm emission filter. The measurement window was localized at some distance from the loading site, allowing the selective recording of loaded mossy fiber with a high signal to noise ratio. The basal fluorescence ($F$) and the amplitude of the fluorescence peak after mossy fiber stimulation ($\Delta F$), and the $\Delta F/F$ ratio were measured in real time. Each recoding was then analyzed individually using Microsoft Excel before pooling them all together. Statistical significance was assessed by either an unpaired Student's $t$-test or a Welch's t. The similarity of variance between each group of results was tested using Ficher's test with $\alpha = 0.02$. ($n$) indicates the number of cells included in the statistics.

**Contextual fear memory**. Animal experiments were approved by the French Agriculture and Forestry Ministry for handling animals (C34-172-13).

Mice (C57Bl6/J, male 8–10 weeks old at the time of surgery, Charles River) were bilaterally implanted with an infusion cannulae (26 gauge, 2.5 mm, Plastics One, Roanoke, VA, USA) aimed at the dorsal hippocampal CA3 using flat skull coordinates: AP: −1.6 mm, ML: ± 2.5 mm, DV: −1.5 mm. The cannulae were fixed to the skull using anchor screws and acrylic dental cement (AgnTho's, Lidingö, Sweden). Following surgery, mice were placed on a heating mat and a dummy cannula was inserted into each guide cannula to seal off the opening. Mice were allowed to recover from surgery for a minimum of one week during which time they were handled and habituated to the drug infusion procedure on a daily basis.

The Pavlovian fear conditioning was performed in a conditioning box (20 cm width × 20 cm length × 20 cm height) placed within a sound proof chamber (Panlab, Barcelona, Spain). Different contexts were used: (A) white walls, metal grid on black floor, washed with 1% acetic acid, or (B) black walls, white rubber floor, washed with 70% ethanol. Mice were conditioned in context A. After 2 min habituation, mice received three pairings (60–120 s variable pairing interval) of a conditioned stimulus (CS: 4 kHz, 80 dB, 30 s tone) with an unconditioned stimulus (US: 2 s, 0.6 mA scrambled footshock) using a freezing system (Panlab). After 24 h, contextual fear was tested by placing the mice in context A for 5 min and after another 24 hours, cued fear was tested by first placing the mice in context B for 2 min and after which the CS was presented twice (120 s intertrial interval). Freezing was measured using a load cell coupler (Panlab) and was defined as the lack of activity above a body weight-corrected threshold for a duration of 1 s or more as analyzed using Freezing software (Panlab).

Drug infusions were made using an injection cannula (33 gauge, 3.5 mm, Plastics One). Immediately following fear conditioning, mice were gently scruffed and an injection cannula was inserted into each guide cannula. The injection cannulae were designed to protrude 1.0 mm from the tip of the guide cannula and effectively penetrated into the hippocampal CA3. Drugs or vehicle were infused at a flow rate of 0.10 µl per min and in a total volume of 0.25 µl per infusion site. Following infusion, the injection cannula was left in place for 1 min to allow drugs to diffuse from the cannula tip. Dummy cannulae were then inserted into each of the guide cannula and mice were returned to the home cage. At the end of each experiment, correct implantation of the guide cannulae was histologically verified on 40 µm slices obtained from brains fixed in 4% paraformaldehyde.

For sample-size estimation using power analyses, an on-line power analysis calculator (G*power3). For each analysis, sample size was determined using a power >0.9 and alpha error = 0.05. All tests were two sided. Parameters for contextual freezing levels and the biological effect of the pharmacological manipulation were based on a previous study[39]. Assuming a mean value of 50% freezing for control mice, and 25% freezing for treated mice, with a standard deviation of 12.5% (which are realistic estimates), than a total sample size of $n = 14$ mice is needed to reject the null hypothesis (two-tailed $t$-test) with 90% probability. This criterion was met for all statistical comparisons.

Mice were randomly assigned to color-coded groups which were subsequently matched to pharmacological treatments. Determination of freezing levels was automated using Freeze software (Panlab, Barcelona, Spain). Cannula implantations were evaluated blinded to pharmacological treatment.

The Kolmogorov—Smirnov normality test was first performed on the data to determine whether parametric or non-parametric tests were required. The data for

freezing levels had a normal distribution and parametric analysis was used. There was no significant difference between the variance of the experimental groups.

**Data availability**. All relevant data are available from the authors and materials will be made available on request.

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

## Acknowledgements

We thank L. Prézeau, D. Maurel, and C. Vol from ARPEGE (Pharmacology Screening-Interactome) facility at the Institut de Génomique Fonctionnelle (Montpellier, France) for their help in various microplate assays. We wish to thank Dr Ilse Smolders and Ria Berckmans for glutamate concentration determination (Vrije Universiteit Brussel, Belgium). Funding was provided by the Centre National de la Recherche Scientifique (CNRS), the Institut National de la Santé et de la Recherche Médicale (INSERM), the University of Montpellier, Cisbio Bioassays, the Fondation Recherche Médicale (FRM DEQ20130326522) and the Fondation Bettencourt Schueller to J.-P.P.; the Fond Unique Interministériel of the french government (FUI, Cell2Lead project) to G.M., J.-P.P. and D.B.; the Agence Nationale de la Recherche (ANR-15-CE18-0020-01) and the labex MabImprove (ANR-10-LABX-5301) to P.R.; the European Union's Seventh Framework Program for research, technological development and demonstration (grant agreement no 627227) to T.C.M.

## Author contributions

J.-P.P., P.R., P.C., D.B., E.V., H.D., H.M.L., E.T., G.M. and E.D. designed the research. D.N. generated the nanobody phage display library and performed the screening and primary characterization of the nanobodies, D.E.M. set up the conditions and prepared the purified mGlu2 receptor in nanodiscs, P.S., D.M.-D., M.M. and E.B. performed the in vitro characterization of the nanobodies, T.C.M. generated the microscopy data, X.R. performed the in silico studies, S.B. performed the experiments on hippocampal slices, D.d.B. performed the in vivo experiments. J.-P.P., P.R. and P.S. wrote the paper with inputs from H.M., P.C. and E.V.

## Additional information

**Competing interests:** The authors declare competing financial interests. Cisbio Bioassays is the manufacturer and the provider of most of the SNAP-tag reagents used in this study. Nanobodies DN1, DN10 and DN13 and their use are protected through the patent WO2016001417 A1.

