## [Peer Review File · Nature Communications]

Reviewers' comments:

Reviewer #1 (Remarks to the Author):

This is an interesting study of high relevance for the field of GPCR research since it identifies allosteric nanobodies that are conformationally-state selective (DN10 and DN13) for the mGlu2 receptor. Although synthetic small molecule PAMs have been described for this and many other GPCRs, the concept of allosteric GPCR anti/nanobodies is newer and relatively unexplored and thus worthy of pursuit with the potential for high impact applications. Overall, I feel that the identification of nanobody PAMs outlined in this study is appropriate for consideration in Nature Communications, but have a number of questions for the authors' consideration that reflect some confusion around the presentation and interpretation of some of their data.

1. The authors need to be consistent and make it much clearer in the multi-panel Figure legends (Main and Supp) as to which experiments were performed in the presence of ambient glutamate and which were performed in cells expressing EAAC1 (i.e., low glutamate).
2. On a related note, it is not clear to me about the relationship between the saturation data shown in Fig. 1 (which state "mean +/- sem of three individual experiments each performed in triplicate") and the quantification of these experiments shown in Table Supp 1, where it is stated that the values are "means +/- sem of n independent determinations"; with n values ranging from 5 to 12. This style of expressing data numbers is ambiguous and needs clarification throughout the manuscript.
3. Why have the authors used a forced coupling of a Gi/o coupled receptor to Gq signaling for functional validation? Does this have the potential to confound interpretation of the native tissue data, which presumably is mediated by Gi/o proteins? This should at least be Discussed. The retention of PAM effects in native tissue at least should mitigate this.
4. DN10 is clearly an ago-PAM but is not pursued any further. Why is this the case? Would the authors expect a better or worse profile in native tissues and/or in vivo?
5. On a related note, the competition data between DN10 and DN13 are used to suggest that they bind to overlapping epitopes, but there are two issues here. Firstly, the potencies in these assays appear much lower than the affinities estimated from the saturation experiments. What is a possible explanation for this? Secondly, and more importantly, the modeling and mutational mapping shown in Fig 3 focus on DN13, but Supp Table 2 also shows the effects on DN10. This is never even mentioned, nor is the fact that DN10 has marked differences in its properties at the mGlu2/3 mutants relative to DN13! This should be addressed, as it may bring into question the claimed specificity of action solely at mGlu2 homodimers or perhaps suggest that DN10 may have a different specificity/selectivity profile. Also, does Fig. 3b imply that two nanobodies are binding per mGlu2 dimer?
6. Minor comment. The authors flip between "context fear" and "contextual fear". The latter

is the correct term.

Reviewer #2 (Remarks to the Author):

The authors generated nanobodies against mGlu2 by performing a phage display selection on a library generated from VHH fragments encoding sequences that were amplified from llamas injected with HEK293 cells expressing mGlu2. mGlu2 expressed on nanodiscs was used as the bait for the selection. To investigate binding of the nanobodies to mGlu2, FRET between a fluorophore attached to the nanobody and a second attached to the receptor was measured. Three nanobodies were reported – two that bind preferentially to activated forms of the mGlu2 and one that binds to inactive receptors as well activated receptors. Using a biosensor that detects movements of the N-termini of the receptors, which correlate with receptor activation, it was shown that the two nanobodies activate mGlu2 in the high nanomolar range in the presence of the mGlu2 agonist LY379268 at EC20. They showed that both DN10 and DN13 potentiate the agonist effect of LY379268. Furthermore, they corroborate these results using an Inositol phosphate accumulation assay. Using a series of deletion mutants of mGlu2 and chimeras of mGlu2 and mGlu3 it was demonstrated that DN 10 and DN 13 bind in the VFT region. Furthermore, DN10 and DN13 did not bind to the mGlu2/4 heteromer. DN13 potentiates the inhibitory effect of a mGlu2/3 agonist on mossy fiber terminals in CA3 of the hippocampus in slices. DN13 injection into the CA3 region potentiated the inhibitory effect of DCG IV on contextual fear conditioning memory consolidation, but not on cued fear conditioning memory consolidation. This result is consistent with mGlu2 being necessary for contextual fear conditioning.

The development of a recombinant PAM that specifically potentiates activated mGlu2 could be extremely useful in a number of contexts, and could ultimately lead to the development of therapeutics. However, it is difficult to determine whether DN13 has these properties because the assays used here to test the nanobody are indirect and difficult to evaluate. In order to give a convincing demonstration that the nanobodies do, in fact, have the properties that the authors claim, it will be necessary to perform standard assays for evaluating affinity reagents in vitro. Furthermore, they must add appropriate controls to experiments in neurons.

They must:

1. Express mGlu2 in cells and provide images that show binding of labeled nanobodies to the cells when they are activated with glutamate (DN10 and DN 13) and when they are not (DN1).
2. Express other (non-mGlu2) mGlu receptors, including chimeras, independently verify that they are on the surface of the cell and show that DN1, DN10 and DN13 don't label these cells.
3. Show that DN1, DN10 and DN13 can pull down mGlu2, but not other mGlu receptors, under appropriate conditions.
4. Provide a full description, as well as a positive control, for the inositol phosphate assay.
5. Provide a full description of how the HTRF ratio is measured.
6. Provide a negative control for the experiments in Figure 4 by replacing DN13 with a

nanobody that does not bind to mGlu2 and show that it has no effect on the physiological state of hippocampal cells or on the behavior of a mouse into which it is injected.

7. Show that application of DN13 to cells in slices either from an mGlu2 knockout mouse, or that have had mGlu2 knocked down, do not show any effects.

Reviewer #3 (Remarks to the Author):

This nice manuscript describes for the first time single-domain antibodies directed against a specific subtype of metabotropic glutamate receptors (the mGlu2 receptor). Two nanobodies, named DN10 and DN13, were shown to act as mGlu2 PAMs, with DN10 also exhibiting intrinsic efficacy (agonist/PAM). This is the first example of nanobodies acting as PAMs of any GPCR. The Authors show that DN13 interacts with both VFTs of homodimeric mGlu2 receptors and can differentiate between mGlu2 homodimers and mGlu2-mGlu4 heterodimers. Thus, nanobodies may become valuable tools for the study of the role played by mGlu2 receptor homodimers vs. heterodimers in physiology and pathology and can also be used for the experimental treatment of CNS disorders involving mGlu2 receptors. The manuscript is extremely interesting and technically sound. This is not surprising considering the scientific reputation of the two corresponding Authors. I have some comments that the Authors may wish to address.

1. If available, extracellular glutamate concentrations after transfection with EACC1 should be reported. This will allow a more direct definition of "low glutamate levels" in relation to the EC50 value of glutamate at mGlu2 receptors.
2. I am surprised that DN13 alone has no effect on its own on presynaptic evoked calcium transients and contextual fear conditioning because presynaptic mGlu2 receptors are activated by the endogenous glutamate (presumably of glial origin). Please, comment.
3. Data obtained in brain slices and in vivo data obtained with intracerebral infusion of DN13 are very convincing. However, the manuscript will be strengthened if the Authors can show that DN13 is functional after systemic administration. Basic VHH (Li et al., 2012) or nanobodies formulated into liposomes might cross the blood-brain barrier. It will be nice to see a behavioral experiment (contextual fear conditioning or others) in which systemic DN13 is tested alone or in combination with subthreshold doses of LY379268 (or any other brain permeable agonist).
4. 3) Line 225: "mGlu2 agonists...act to consolidate contextual fear memory". Please, correct.
5. Lines 228-231: I disagree that potentiation of DGC-IV response by DN13 indicates that only mGlu2 receptors are involved. mGlu2 and mGlu3 may have redundant functions there and also form heterodimers (in addition to homodimers). The use of subtype-selective NAMs (rather than PAMs) may help to establish whether mGlu3 have a role. Potentiation by DN13 indicates that mGlu2 homodimers are certainly involved.

Reviewer #4 (Remarks to the Author):

This manuscript by Scholler et al. reports development and verification of nanobodies

functioning as a positive allosteric modulator of mGlu2 homodimer. These mGlu2 nanobodies work in the nanomolar range and bind only to the active form of mGlu2. One of the two nanobodies (DN13) does not have intrinsic agonist activity, and, importantly, potentiates mGlu2 activity in brain slices and in vivo, increasing contextual but not cued fear conditioning by CA3 drug infusion. This is the first development of nanobodies for GPCRs, and mGluR2 is an important target for both basic research and clinical application. In addition, the experiments were well designed, and the results are largely convincing.

Major comments:

1. Although the contrasting actions of DN13 on contextual and cued fear conditioning are impressive, I wonder whether the authors have tested basic behaviors such as locomotion or anxiety, which would affect the freezing rate.
2. Is there any reason why the authors did not test mGlu3 mutB in Figure 3e? Given that mut B has significant impacts on the DN13 binding, testing whether mut B is sufficient to confer the binding to mGlu3 would be important.

Minor comments:

1. What is the difference between HTRF ratio in Figure 1b and normalized HTRF ratio in Figure 1f?
2. Is there any reason why the numbers in the y axis of Figure 1f and Figure 3d are so different?
3. It would help readers if the authors could label which are key residues important for nanobody binding (A248K, mut A, and mut B) in Figure 3c or somewhere.
4. In Figure 4c, is there any reason why the authors did not try drug infusion before conditioning or immediately before contextual/cue test?

Answers to the Reviewers' comments:

Reviewer #1 (Remarks to the Author):

This is an interesting study of high relevance for the field of GPCR research since it identifies allosteric nanobodies that are conformationally-state selective (DN10 and DN13) for the mGlu2 receptor. Although synthetic small molecule PAMs have been described for this and many other GPCRs, the concept of allosteric GPCR anti/nanobodies is newer and relatively unexplored and thus worthy of pursuit with the potential for high impact applications. Overall, I feel that the identification of nanobody PAMs outlined in this study is appropriate for consideration in Nature Communications, but have a number of questions for the authors' consideration that reflect some confusion around the presentation and interpretation of some of their data.

1. The authors need to be consistent and make it much clearer in the multi-panel Figure legends (Main and Supp) as to which experiments were performed in the presence of ambient glutamate and which were performed in cells expressing EAAC1 (i.e., low glutamate).

This information is now clearly stated in the figures, the figure legends and the materials and methods section.

2. On a related note, it is not clear to me about the relationship between the saturation data shown in Fig. 1 (which state "mean +/- sem of three individual experiments each performed in triplicate") and the quantification of these experiments shown in Table Supp 1, where it is stated that the values are "means +/- sem of n independent determinations"; with n values ranging from 5 to 12. This style of expressing data numbers is ambiguous and needs clarification throughout the manuscript.

Data shown in Fig.1 correspond to the means of three experiments performed in triplicates. More experiments have been performed thereafter as controls of other experiments, and the means of the Kd values determined from all experiments are indicated in Table 1, with the exact number of determinations (n). The referee can notice that the means Kd values in Table 1 nicely fit with the data shown in Figure 1.

We went through the entire manuscript to verify that the n indicated corresponds to the data shown.

3. Why have the authors used a forced coupling of a Gi/o coupled receptor to Gq signaling for functional validation? Does this have the potential to confound interpretation of the native tissue data, which presumably is mediated by Gi/o proteins? This should at least be Discussed. The retention of PAM effects in native tissue at least should mitigate this.

As reported in most of our studies with group-II and group-III mGluRs, we found it much easier to measure G protein activation using a chimeric G protein enabling these receptors to couple to the PLC pathway. Indeed quantifying the PLC activation is much easier, especially for transient transfected cells, than the quantification of the inhibition of forskolin-induced cAMP formation. Indeed, while the inhibition occurs only in the transfected cells, forskolin activates adenylyl cyclase in every cell. However, we agree with the referee that what is observed with this assay may not necessarily apply to the natural coupling of mGlu2 receptors to Gi/o proteins. However, as noticed by the referee, similar data were obtained in transfected cells using the Gq β 9/PLC assay, and in brain slices, such as the pure PAM activity of DN13, as well as the ago-PAM activity of DN10 (see the new Figure 5). Whatever, we set up an assay based on the use of the pGlo sensor to analyze Gi coupling of the mGlu2 receptor. In this assay, the plasmid encoding the pGlo sensor was co-transfected with the mGlu2 receptor, such that only the response generated in the transfected cells is measured. As shown in a new panel (panel b) in sup Fig. 3, 100 nM DN13 clearly potentiated the effect of the mGlu2 agonist LY379268 in inhibiting forskolin-induced cAMP formation, consistent with what was observed using the chimeric G protein. New figures (new Figure 5, and sup

Fig 3b) have been added, and the text has been modified to clarify this issue.

4. DN10 is clearly an ago-PAM but is not pursued any further. Why is this the case? Would the authors expect a better or worse profile in native tissues and/or in vivo?

We did perform experiments with DN10 in hippocampal slices, and these are now included in the revised Fig. 5 (ex Figure 4). These data confirmed that DN10 is an ago-PAM, since a direct slowly developing inhibition of pre-synaptic Ca^{2+} signal can be detected with DN10, in the absence of DCG-IV. After a saturating concentration of DCG-IV is applied, the recovery after washing is much slower in the presence of DN10, in agreement with its PAM activity. Experiments with living animals are highly controlled through ethic committees, with the aim to limit the number of animals to be tested. Accordingly we had to justify why to do twice as much experiments with two different nanobodies. It was not obvious, for a first set of experiments to justify this, unless one wanted to specifically examine the effect of an ago-PAM, compared to a pure PAM. Accordingly, we decided to concentrate our experiments on the nanobody offering what we considered as the most interesting properties.

5.a On a related note, the competition data between DN10 and DN13 are used to suggest that they bind to overlapping epitopes, but there are two issues here. Firstly, the potencies in these assays appear much lower than the affinities estimated from the saturation experiments. What is a possible explanation for this?

The competition experiments were performed using d2-labeled DN1, DN10 or DN13. The d2-labeling does not modify the K_d of DN1 (4.5 ± 0.5 nM ($n=3$)) and DN10 (3.3 ± 0.5 nM ($n=3$)), but affects that of DN13 (61 ± 14 nM ($n=5$)) instead of 3.5 nM for the unlabeled DN13. According to the concentration of d2-labeled nanobody used (20 nM for DN1-d2, and 75 nM for DN10-d2 and DN13-d2), and the IC_{50} measured, one could determine, according to the equation $IC_{50} = K_i(1 + C/K_d)$ corresponding to a competitive inhibition, K_i values for DN1, DN10 and DN13 against their d2 equivalent of: 4.9, 2.2 and 5.0 nM respectively, and for DN10 in inhibiting DN13-d2: 4.3, and DN13 inhibiting DN10-d2: 1.0, in line with the K_d determined for these 3 nanobodies in saturation assays. This is now clearly stated in the legend to this supplementary figure of the revised version of our manuscript.

5.b Secondly, and more importantly, the modeling and mutational mapping shown in Fig 3 focus on DN13, but Supp Table 2 (This is indeed Supp Table 3) also shows the effects on DN10. This is never even mentioned, nor is the fact that DN10 has marked differences in its properties at the mGlu2/3 mutants relative to DN13! This should be addressed, as it may bring into question the claimed specificity of action solely at mGlu2 homodimers or perhaps suggest that DN10 may have a different specificity/selectivity profile.

The data obtained with DN10 are now mentioned in the text, and it is clearly stated that DN10 and DN13 do not recognize exactly the same epitope even though they compete, suggesting overlapping epitopes, or steric hindrance between these two nanobodies. We performed additional experiments with the three nanobodies (DN1, DN10 and DN13) on both mGlu2-3 and mGlu2-4 heterodimers (see new Figure 4, and the new sup Figure 8). First binding experiments were performed in cells expressing mGlu2 wild type, and a SNAP version of either mGlu3 or mGlu4 labeled with Lumi4-Tb. Accordingly, binding of the nanobodies could not generate TR-FRET signal with either mGlu2 homodimers (not labeled with the donor) nor with mGlu3 or 4 homodimers since the nanobodies do not bind on these homodimers. Accordingly, any TR-FRET signal is expected to originate from the binding of the nanobodies on the mGlu2-3 or 2-4 heterodimers. Using this approach, binding of DN1 was clear on both heterodimers (new Figure 4), as expected due to the binding of this nanobody on the mGlu2 subunit whatever the conformation, then unlikely to interact at the interface between the subunit. No binding of DN10 and DN13 could be detected on these heterodimers in the presence of antagonist (new Figure 4). However, a clear signal saturating to values 4 times lower than those obtained with DN1 was obtained with both mGlu2-3 and mGlu2-4. The low saturated TR-FRET signal suggests that DN10 and DN13 bind to a fraction of cell surface receptors carrying the SNAP-tag, a fraction smaller than that

of receptors interacting with DN1. Because under our experimental conditions cells express at the cell surface not only the heterodimer, but also the homodimers, we think that the low TR-FRET signal originates from the presence of possible mGlu2-2 + mGlu2-3 oligomers. Indeed, mGluR complexes larger than dimers can be detected, especially when stabilized in an active form (Xue et al., Nat Chem Biol 2016 for mGlu5 oligomers, and our unpublished data for mGlu2 oligomers) (new Figure 4).

Second, we examined whether the nanobodies could affect the function of either mGlu2-3 or mGlu2-4 heterodimers using the TR-FRET sensor assay. Consistent with the proposed interpretation of the binding data, DN10 and DN13 had no significant effect on either heterodimer (new Figure 4 and new supplementary Figure 8).

These data confirm that both DN10 and DN13 interact with both subunits of the mGlu2 homodimer. Although key residues from protomer B were clearly identified (Figure 3), those of protomer A were not, as the mutations introduced in mGlu2 did not affect DN binding. More work will be necessary to clarify this issue.

Taken together, these data revealed that DN10 and DN13 are specific for mGlu2 homodimers in a nanomolar concentration range. Our data are then consistent with the effect of mGlu2/3 agonists at the mossy fiber terminals in the hippocampus being mainly mediated by mGlu2 homodimers. This is now clearly stated in our revised manuscript.

5.c Also, does Fig. 3b imply that two nanobodies are binding per mGlu2 dimer?

This is a very good point. This is what one would predict because of the presence of two binding epitopes in an mGlu2 homodimer. We indeed verified this using a FRET based assay using d2-labeled nanobodies, and Lumi4-Tb-labeled nanobodies. We found a large TR-FRET signal using the DN1 nanobody, of similar amplitude in the presence of agonist or antagonist, as expected because the binding of DN1 is independent of the conformational state of mGlu2. In the case of DN10 and DN13, no signal could be detected over that measured in mock transfected cells in the presence of antagonist, but a clear signal was obtained when the receptor is activated indicating that mGlu2 activation allows the binding of two DN10 or DN13 nanobodies in close proximity. As expected because of the lower affinity of d2-labeled DN13, the TR-FRET signal measured with this nanobody in the presence of agonist is lower than that measured with DN10. These data have been added in a new supplementary figure (new sup Figure 7).

6. Minor comment. The authors flip between “context fear” and “contextual fear”. The latter is the correct term.

This has been corrected throughout the manuscript.

Reviewer #2 (Remarks to the Author):

The authors generated nanobodies against mGlu2 by performing a phage display selection on a library generated from VHH fragments encoding sequences that were amplified from llamas injected with HEK293 cells expressing mGlu2. mGlu2 expressed on nanodiscs was used as the bait for the selection. To investigate binding of the nanobodies to mGlu2, FRET between a fluorophore attached to the nanobody and a second attached to the receptor was measured. Three nanobodies were reported – two that bind preferentially to activated forms of the mGlu2 and one that binds to inactive receptors as well activated receptors. Using a biosensor that detects movements of the N-termini of the receptors, which correlate with receptor activation, it was shown that the two nanobodies activate mGlu2 in the high nanomolar range in the presence of the mGlu2 agonist LY379268 at EC20. They showed that both DN10 and DN13 potentiate the agonist effect of LY379268. Furthermore, they corroborate these results using an Inositol phosphate accumulation assay. Using a series of deletion mutants of mGlu2 and chimeras of mGlu2 and mGlu3 it was demonstrated that DN

10 and DN 13 bind in the VFT region. Furthermore, DN10 and DN13 did not bind to the mGlu2/4 heteromer. DN13 potentiates the inhibitory effect of a mGlu2/3 agonist on mossy fiber terminals in CA3 of the hippocampus in slices. DN13 injection into the CA3 region potentiated the inhibitory effect of DCG IV on contextual fear conditioning memory consolidation, but not on cued fear conditioning memory consolidation. This result is consistent with mGlu2 being necessary for contextual fear conditioning.

The development of a recombinant PAM that specifically potentiates activated mGlu2 could be extremely useful in a number of contexts, and could ultimately lead to the development of therapeutics. However, it is difficult to determine whether DN13 has these properties because the assays used here to test the nanobody are indirect and difficult to evaluate. In order to give a convincing demonstration that the nanobodies do, in fact, have the properties that the authors claim, it will be necessary to perform standard assays for evaluating affinity reagents *in vitro*. Furthermore, they must add appropriate controls to experiments in neurons.

In our study we determined the nanobodies affinities using a TR-FRET-based approach, well validated for the characterization of ligand interaction to their receptor (see for example Hansou et al Durroux (2014) ACS Chem Biol and Herenbrink et al., Nat Commun 2016). The main advantage is that it overcomes the difficulties related to the use of *in vitro* assays such as Biacore, since the affinities are determined on the target protein in its cellular environment. We think this approach is then more relevant for *in cellulo* and *in vivo* experiments. We are not convinced that the use of other approaches will bring much to our study.

They must:

1. Express mGlu2 in cells and provide images that show binding of labeled nanobodies to the cells when they are activated with glutamate (DN10 and DN 13) and when they are not (DN1).

We now provide images using fluorescently labeled nanobodies on HEK 293 cells expressing mGlu2 or mGlu3 (as a negative control), and show that the labeling is agonist dependent for DN13, and not for DN1. We also performed experiments using both mGlu2- and mGlu4-transfected hippocampal neurons with similar data. The data with DN13 on mGlu2 transfected neurons under control condition, or after addition of either the agonist LY379268 (150 nM), or the antagonist LY341495 (1 μ M) have been included in the **revised Figure 1 (panel g)**. The data with DN1 on HEK cells expressing either mGlu2 or mGlu3, and in transfected neurons expressing either mGlu2 or mGlu4 have been added in the **new sup Figure 2, panels a and b**, respectively. The data obtained with mGlu2 expressing HEK 293 cells are in **new sup Figure 2, panels c**.

2. Express other (non-mGlu2) mGlu receptors, including chimeras, independently verify that they are on the surface of the cell and show that DN1, DN10 and DN13 don't label these cells.

Binding of the DN1, DN10 and DN13 to the eight full-length wild-type rat mGlu receptors was already shown in Supp Fig. 1. No binding was observed on any mGlu other than mGlu2 receptors both in the presence or absence of ligands. The absence of binding to the non-mGlu2 receptors was not due to the loss of cell surface expression of these receptors, since they were all strongly detected at the cell surface (Supp Fig.1d). As indicated in our answer to point 1, cells expressing the other mGluR (mGlu3 or mGlu4) have been used as negative controls in a cell imaging assay, and again confirm cell labeling only when mGlu2 is expressed (**new sup Figure 2, panels a and b**). Of note, the mGlu2 labeling with DN13 is highly dependent on the presence of agonist, no labeling being observed in the absence of ligand or in the presence of antagonist (**new Figure 1g, new sup Figure 2c**).

3. Show that DN1, DN10 and DN13 can pull down mGlu2, but not other mGlu receptors, under appropriate conditions.

We think our original data, plus those now included with cell imaging, including neurons, demonstrate the specificity of the three nanobodies. The aim of developing these nanobodies was not to conduct biochemical studies, but rather functional studies. Accordingly, we don't think pull down assays will bring much to the present study, and we preferred to concentrate on other more important issues during the revision of our manuscript.

4. Provide a full description, as well as a positive control, for the inositol phosphate assay. Such information has been added in the materials and methods section. Of note the protocol is a standard protocol and follows the manufacturer instruction. This assay is being routinely used in many laboratories nowadays, so we wonder what the referee refers to when he/she is asking for a positive control, since much can be found on the Cisbio Bioassays web site. In our hands, Gq-coupled mGluRs do generate IP₁, leading to a change in signal in the IP One assay (Trinquet et al., *Anal Biochem* 2006), while Gi-coupled mGluRs do not, unless co-expressed with a chimeric G protein allowing their coupling to PLC (for example, see Xue et al. *Nat Chem Biol* 2015).

5. Provide a full description of how the HTRF ratio is measured. Such information has been added in the materials and methods section. Much details on the assay can be found in Scholler et al., *Nat Chem Biol* 2017, and Moreno-Delgado et al. *eLife* 2017).

6. Provide a negative control for the experiments in Figure 4 by replacing DN13 with a nanobody that does not bind to mGlu2 and show that it has no effect on the physiological state of hippocampal cells or on the behavior of a mouse into which it is injected. DN13 has no effect on its own, and only potentiates the effect of DCG-IV when used at low concentration. When using saturating concentrations of DCG-IV, DN13 did not increase the maximal effect observed, but significantly slows down the recovery after washing, in agreement with the PAM effect of this nanobody on mGlu2. We now added more data showing that the long-term effect observed after DN13 and DCG-IV application can be rapidly inhibited with the group-II mGlu antagonist LY341495 (new Sup Fig. 9). Moreover, we show that DN10 that display an ago-PAM activity at mGlu2 in transfected HEK cells, also display an ago-PAM activity in these brain slices (new Fig. 5). Eventually, we now show that DN1 that binds on mGlu2 but has no functional effect in transfected cells expressing mGlu2, is devoid of activity in brain slices, displaying no detectable activity at the mossy fiber terminals when applied alone, and have no potentiating activity of the DCG-IV response (new Sup Fig. 9). We think these data clearly demonstrate that the observed effect involves action of DN13 at mGlu2 receptors.

7. Show that application of DN13 to cells in slices either from an mGlu2 knockout mouse, or that have had mGlu2 knocked down, do not show any effects. We do not have access to mGlu2 KO mice. The inclusion of such experiments will require the generation of this mouse line in our animal facility, and this will need at least a year. Although we agree with the referee that such control would strengthen the in vivo data, we still think that when put in parallel with the data obtained in the same brain area in slices, plus the fact that DN13 has no effect on its own, but only potentiated the effect of DCG-IV, bring strong evidence for the involvement of mGlu2 receptors in the observed effect.

Reviewer #3 (Remarks to the Author):

This nice manuscript describes for the first time single-domain antibodies directed against a specific subtype of metabotropic glutamate receptors (the mGlu2 receptor). Two nanobodies, named DN10 and DN13, were shown to act as mGlu2 PAMs, with DN10 also exhibiting intrinsic efficacy (agonist/PAM). This is the first example of nanobodies acting as PAMs of any GPCR. The Authors show that DN13 interacts with both VFTs of homodimeric mGlu2 receptors and can differentiate between mGlu2 homodimers and mGlu2-mGlu4

heterodimers. Thus, nanobodies may become valuable tools for the study of the role played by mGlu2 receptor homodimers vs. heterodimers in physiology and pathology and can also be used for the experimental treatment of CNS disorders involving mGlu2 receptors. The manuscript is extremely interesting and technically sound. This is not surprising considering the scientific reputation of the two corresponding Authors. I have some comments that the Authors may wish to address.

1. If available, extracellular glutamate concentrations after transfection with EAAC1 should be reported. This will allow a more direct definition of "low glutamate levels" in relation to the EC50 value of glutamate at mGlu2 receptors.

The reported glutamate affinity at mGlu2 receptors is 12.3 μM (Johnson et al., 2000). In our hands, by displacing bound [^3H]-LY341495, we measured a K_i for glutamate of 9.5 μM (unpublished data). The glutamate EC50 in control transfected cells is around 10 μM when determined using the Gq β 9 inositol phosphate assay (Gomez et al., Mol Pharmacol 1996; Brabet et al., Neuropharmacology 1998; Parmentier et al., Neuropharmacology 2000). The glutamate concentration in the cell medium was found to be $2.7 \pm 0.2 \mu\text{M}$ (n=4) in the absence of the glutamate transporter EAAC1, and $0.75 \pm 0.07 \mu\text{M}$ (n=5) when cells expressed this transporter. We still think the latter value is over-estimated, the glutamate concentration being very likely lower in the vicinity of the cells, then in the receptor close environment. Of note, the measured glutamate EC50 on mGlu2 increased from 10 to 100 μM in the presence of EAAC1 (see Doumazane et al. PNAS 2013), demonstrating the efficient capacity of the transporter to lower the glutamate concentration on the vicinity of the receptor. We assumed this low apparent EC50 results from the large decrease in the real glutamate concentration close to the cell surface due to the efficient uptake of extracellular glutamate, since no change in potency was observed with other mGluR ligands not taken up by EAAC1.

2. I am surprised that DN13 alone has no effect on its own on presynaptic evoked calcium transients and contextual fear conditioning because presynaptic mGlu2 receptors are activated by the endogenous glutamate (presumably of glial origin). Please, comment.

The extracellular concentration of glutamate in the brain is tightly regulated, in contrast to the situation at the periphery and is maintained as much as possible to value in the high nanomolar range. This is mainly due to the presence of high affinity glutamate transporters in astrocytes and in neurons, but absent at the periphery. Such transporters are also absent from commonly used cell lines (such as HEK 293 cells used in this study), such that the extracellular concentration in the culture medium is not at all controlled, and can reach up to 100 μM .

However, it is true that the mGlu2/3 antagonist LY341495 displays effects in the brain, including cFos expression in a specific subset of neurons, with behavioral consequences (Linden et al., Neuropharmacology 2005). However, whether this is due to mGlu2, or to mGlu3 that displays a 10-20 fold higher affinity for glutamate is not clear to me.

In the slice preparation that we used, the medium is constantly changed (2 ml/min), then limiting any possible increase in glutamate concentration due to neuronal activity. This is well illustrated by the absence of effect of the mGlu2 PAM LY487379 (data not shown).

Moreover, as now shown in new Fig. 5, in contrast to DN13, DN10 that displays agonist activity is indeed able to decrease Ca^{2+} signals in the mossy fiber terminals, in agreement with its ago-PAM activity. Eventually, the mGlu2 antagonist LY341495 also has no effect on its own, indicating that there is no significant basal activation of mGlu2 in the slice preparation. The situation may be different *in vivo* where there is no constant washing of the area where the nanobody is injected. However, our data are clear, indicating that there not enough endogenous glutamate in this area to allow DN13 to have an effect alone resulting from the potentiation of the ambient glutamate-mediated response.

3. Data obtained in brain slices and *in vivo* data obtained with intracerebral infusion of DN13 are very convincing. However, the manuscript will be strengthened if the Authors can show

that DN13 is functional after systemic administration. Basic VHH (Li et al., 2012) or nanobodies formulated into liposomes might cross the blood-brain barrier. It will be nice to see a behavioral experiment (contextual fear conditioning or others) in which systemic DN13 is tested alone or in combination with subthreshold doses of LY379268 (or any other brain premeable agonist).

This is our dream ! However, nanobodies do not easily pass the blood brain barrier (Rissiek et al., *Frontiers Cell Neurosci* 2014; Caljon et al., *BJP* 2012). There are several strategies like those indicated by the referee that can be used and we are actually working on these. Based on our preliminary data, this will take some time and we are not sure to succeed in a reasonable period of time. We then think this is out of the scope of the present manuscript.

4. 3) Line 225: "mGlu2 agonists...act to consolidate contextual fear memory". Please, correct.

The sentence has been corrected.

5. Lines 228-231: I disagree that potentiation of DGC-IV response by DN13 indicates that only mGlu2 receptors are involved. mGlu2 and mGlu3 may have redundant functions there and also form heterodimers (in addition to homodimers). The use of subtype-selective NAMs (rather than PAMs) may help to establish whether mGlu3 have a role. Potentiation by DN13 indicates that mGlu2 homodimers are certainly involved.

As indicated above (our answer to referee 1, point 5b), more data are now provided on the effect of DN10 and DN13 of both mGlu2-3 and mGlu2-4 heterodimers (see new Fig.4 and new sup Fig. 8). We found that at concentration up to 300 nM neither DN10 nor DN13 had any significant functional effect on mGlu2-3 and mGlu2-4 heterodimers. This is now indicated in the revised manuscript.

Reviewer #4 (Remarks to the Author):

This manuscript by Scholler et al. reports development and verification of nanobodies functioning as a positive allosteric modulator of mGlu2 homodimer. These mGlu2 nanobodies work in the nanomolar range and bind only to the active form of mGlu2. One of the two nanobodies (DN13) does not have intrinsic agonist activity, and, importantly, potentiates mGlu2 activity in brain slices and in vivo, increasing contextual but not cued fear conditioning by CA3 drug infusion. This is the first development of nanobodies for GPCRs, and mGluR2 is an important target for both basic research and clinical application. In addition, the experiments were well designed, and the results are largely convincing.

Major comments:

1. Although the contrasting actions of DN13 on contextual and cued fear conditioning are impressive, I wonder whether the authors have tested basic behaviors such as locomotion or anxiety, which would affect the freezing rate.

In our experiments, cannulated mice were used only once for behavioral experiments to avoid potential bias due to repeated local infusion of compounds.

We based our behavioral experiments on a previous study on the effects of dorsal hippocampal administration of the mGlu2 agonist DCG-IV on fear memory processing in mice (Daumas et al. 2015 *Learning & Memory*). In this study, the authors found that administration of DCG-IV into the CA3 either before or after fear conditioning impaired contextual freezing responses without affecting cued freezing. The effects of DCG-IV were found to last up to 3 h after administration (Daumas et al. 2009 *Learning & Memory*). In our study, contextual fear and cued fear were tested one and two days after drug administration. At this time point, direct effects of mGlu2 activation on locomotor activity or anxiety appears highly unlikely.

We acknowledge that systemic administration of mGlu2/3 agonists is anxiolytic and that this effect is associated with decreased c-Fos expression in the hippocampus (Linden et al. 2004 Neuropsychopharmacology). However, it should be noted that we administered compounds in the dorsal hippocampus, which is widely recognized to be involved in contextual information processing, whereas the ventral hippocampus is thought to be implicated in emotion and stress (Fanselow and Dong 2010 Neuron). This makes it further unlikely that the observed effects on freezing were secondary to modulation of anxiety levels.

In the Daumas et al. 2009 Neuropharmacology study, the authors found that administration of 1 nmol DCG-IV before conditioning induced non-specific side effects, leading to reduced locomotor activity and decreased freezing during the cued fear test. However, in our study, cued freezing remained unaffected in all experimental groups, suggesting that similar non-specific side effects did not occur.

2. Is there any reason why the authors did not test mGlu3 mutB in Figure 3e? Given that mut B has significant impacts on the DN13 binding, testing whether mut B is sufficient to confer the binding to mGlu3 would be important.

We did generate first mGlu3 mutant A, and then mutant AB, by adding novel mutations in the mutant A sequence, with the aim to gain function of the nanobody on the mGlu3 receptor. The aim was not to go into details of the role of each residues in DN13 binding, but to prove that the proposed epitope was correct by recreating it in a receptor that do not normally bind DN13. The text has been modified to clarify this issue, and especially the involvement of mutated protomer B residues, but not those of protomer A.

Minor comments:

1. What is the difference between HTRF ratio in Figure 1b and normalized HTRF ratio in Figure 1f?

HTRF ratio is calculated as described in the materials and methods section (sensitized acceptor emission at 665 nm / donor emission à 620 nm). Figure 1b shows the means of data obtained from 3 independent experiments. For the binding saturation curves (figure 1d-f), the HTRF ratio over background of each individual experiments were normalized to the maximal signal, and then the means values calculated. The data expressed in % of the maximal signal. This was done on purpose to allow the referee to concentrate on the affinity of the 3 different nanobodies. To clarify this issue, the legend of the Y axis in the revised Figure 1 panels d-f was changed to HTRF ratio (% of max)

2. Is there any reason why the numbers in the y axis of Figure 1f and Figure 3d are so different?

As indicated above, the data in Figure 1f were normalized to the maximal response, while those in Figure 3d were not.

3. It would help readers if the authors could label which are key residues important for nanobody binding (A248K, mut A, and mut B) in Figure 3c or somewhere.

This information has been added in the revised figure.

4. In Figure 4c, is there any reason why the authors did not try drug infusion before conditioning or immediately before contextual/cue test?

As discussed above, the effects of mGlu2 activation prior to conditioning was previously investigated (Daumas et al. 2009). We chose to study the effects of drug infusion immediately after conditioning, given that in this configuration potential bias due to effects on locomotor activity or anxiety was the most unlikely.

REVIEWERS' COMMENTS:

Reviewer #1 (Remarks to the Author):

In the revised MS, the authors have appropriately addressed my main comments.

Reviewer #2 (Remarks to the Author):

Overall I am satisfied that my questions have been answered by the revisions made by the authors to the manuscript and by the explanations provided in the rebuttal letter.

Reviewer #3 (Remarks to the Author):

The Authors have nicely addressed all my requests. I have no further comments. Congratulations for this excellent work.

Reviewer #4 (Remarks to the Author):

The authors have fully addressed my comments, and I do not have any further comments.